Manuscript prepared for Biogeosciences Discuss.
with version 2014/09/16 7.15 Copernicus papers of the LaTeX class copernicus.cls.
Date: 6 March 2017

# Potential sources of variability in mesocosm experiments on the response of phytoplankton to ocean acidification

**Maria Moreno de Castro**[1]**, Markus Schartau**[2]**, and Kai Wirtz**[1]

[1]Helmholtz-Zentrum Geesthacht, Centre for Materials and Coastal Research
[2]GEOMAR Helmholtz Centre for Ocean Research Kiel

Correspondence to: M. M. de Castro (maria.moreno@hzg.de)

## Abstract

Mesocosm experiments on phytoplankton dynamics under high $CO_2$ concentrations mimic the response of marine primary producers to future ocean acidification. However, potential acidification effects can be hindered by the high standard deviation typically found in the
replicates of the same $CO_2$ treatment level. In experiments with multiple unresolved factors and a sub-optimal number of replicates, post-processing statistical inference tools might fail to detect an effect that is present. We propose that in such cases, data-based model analyses might be suitable tools to unearth potential responses to the treatment and identify the uncertainties that could produce the observed variability. As test cases, we used
data from two independent mesocosm experiments. Both experiments showed high standard deviations and, according to statistical inference tools, biomass appeared insensitive to changing $CO_2$ conditions. Conversely, our simulations showed earlier and more intense phytoplankton blooms in modeled replicates at high $CO_2$ concentrations and suggested that uncertainties in average cell size, phytoplankton biomass losses, and initial nutrient
concentration potentially outweigh acidification effects by triggering strong variability during the bloom phase. We also estimated the thresholds below which uncertainties do not escalate to high variability. This information might help in designing future mesocosm experiments and interpreting controversial results on the effect of acidification or other pressures on ecosystem functions.
*Keywords: Variability, Uncertainty Quantification, Mesocosms, Phytoplankton, Ocean Acidification*

## 1 Introduction

Oceans are a sink for about 30% of the excess atmospheric $CO_2$ generated by human activities (Sabine et al., 2004). Increasing carbon dioxide concentration in aquatic environments
alters the balance of chemical reactions and thereby produces acidity, which is known as ocean acidification (OA) (Caldeira and Wickett, 2003). Interestingly, the sensitivity of pho-

toautotrophic production of particulate organic carbon (POC) to OA is less pronounced than previously thought. Several studies on $CO_2$ enrichment revealed an overall increase in POC (e.g. Schluter et al., 2014; Eggers et al., 2014; Zondervan et al., 2001; Riebesell et al., 2000), but other studies did not detect $CO_2$ effects on POC concentration (e.g. Jones et al.,
2014; Engel et al., 2014) or primary production (Nagelkerken and Connell, 2015). General compilation studies that document controversial results are e.g., Riebesell and Tortell (2011) and Gao et al. (2012).

In some experiments, the different treatment levels, i.e., different $CO_2$ concentrations, have been applied in parallel repetitions, also known as replicates or sample units. This
was the case in several $CO_2$ perturbation experiments with mesocosms (Riebesell et al., 2008). Often, high variances are found in measurements among replicates of similar $CO_2$ levels (Paul et al., 2015; Schulz et al., 2008; Engel et al., 2008; Kim et al., 2006; Engel et al., 2005). It is this variance in data that reflects system variability ,thereby introducing a severe reduction in the ratio between a true acidification response signal and the variability
in observations. Ultimately, the experimental data exhibit a low signal-to-noise ratio.

Mesocosms typically enclose natural plankton communities, which is a more realistic experimental setup compared to batch or chemostat experiments with monocultures (Riebesell et al., 2008). Along with this, mesocosms allow for a larger number of possible planktonic interactions that provides opportunities for the spread of uncontrolled heterogeneity.
Moreover, physiological states vary for different phytoplankton cells and environmental conditions. For this reason, independent experimental studies at similar but not identical conditions might yield divergent results. The variability in data of mesocosm experiments is thus generated by variations of ecological details, i.e., small differences among replicates of the same sample, like in species abundance, nutrient concentration and metabolic states of
the algae at the initial setup of the experiments. Differences of these factors often remain unresolved and might therefore be treated as uncertainties in a probabilistic approach.

To account for all possible factors that determine all differences in plankton dynamics is practically infeasible, which also impedes a retrospective statistical analysis of the experimental data. However, since unresolved ecological details might propagate over the course

of the experiment, it is meaningful to consider a dynamical model approach to upgrade the data analysis. From a modeling perspective, some important unresolved factors translate into (i) uncertainties in specifying initial conditions (of the state variables), and (ii) uncertainties in identifying model parameter values. Here, we apply a dynamical model to estimate the effects of ecophysiological uncertainties on the variability in POC concentration of two mesocosm experiments. Our model describes plankton growth in conjunction with a dependency between $CO_2$ utilization and mean logarithmic cell size (Wirtz, 2011). The structure of our model is kept simple, thereby reducing the possibility of overparameterizing the mesocosms dynamics. The model is applied to examine how uncertainties in individual factors, namely initial conditions and parameters, can produce the standard deviation of the distribution of observed replicate data. Our main working hypotheses on the origins of variability in mesocosm experiments are the following:

- differences among replicates of the same sample can be interpreted as unresolved random variations (named uncertainties hereafter)

- uncertainties can amplify during the experiment and generate considerable variability in the response to a given treatment level

- which uncertainties are more relevant can be estimated by the decomposition of the variability in the experimental data.

For our data-supported model analysis of variability decomposition we consider the propagation of distributions (JCGM, 2008b) to seek potential treatment responses that are masked by the variability in observations of two independent OA mesocosm experiments, namely, Pelagic Enrichment $CO_2$ Experiment (PeECE II and III). The central idea is to produce ensembles of model simulations, starting from a range of values for selected factors. The range of values for these selected factors is determined so as the variability in model outputs does not exceed variability in observations over the course of the experiment. The margins of the variational range of each factor were thus confined by the ability of the dynamical model to reproduce the magnitude of the variability observed in POC. These

confidence intervals describe the tolerance thresholds below which uncertainties do not escalate to high variability in the modeled replicates, and can serve as an estimator of the tolerance of experimental replicates to such uncertainties. This information can be important to ensure reproducibility, allowing a comparison between the results of different independent experiments and increasing confidence regarding the effects of OA on phytoplankton (Broadgate et al., 2013).

## 2 Method

Potential sources of variability are estimated following a procedure already applied in system dynamics, experimental physics and engineering JCGM (2008b). The basic principles of uncertainty propagation are summarized here using a six-step method (see Fig. 1). Steps 1 and 2 are described in subsection 2.1 and comprise a classical model calibration (using experimental data of biomass and nutrients) to obtain the reference run representing the mean dynamics of each treatment level. In this way we found the reference value for the model factors, i.e., parameters and initial conditions. Steps 3 and 4, described in subsection 2.2, include the tracked propagation of uncertainties by systematically creating model trajectories for POC, each one with a slightly different value of a model factor. In steps 5 and 6, we estimated the thresholds of the model-generated variability and the effect of the uncertainty propagation (also explained in subsection 2.2).

### 2.1 Model setup, data integration, and description of the reference run

In this section, we describe the biological state that was used as reference dynamics. Our model resolves a minimal set of state variables insofar monitored during experiments that are assumed to be key agents of the biological dynamics. Model equations are shown in Table 1. Reference values of the parameters are shown in Table 2. An exhaustive model documentation is given in Appendix B. The model simulates experimental data from the Pelagic Enrichment $CO_2$ Experiment (PeECE), a set of 9 outdoor mesocosms placed in coastal waters close to Bergen (Norway) during the spring seasons of 2003 (PeECE II) and

2005 (PeECE III). In both the experiments, blooms of the natural phytoplankton community were induced and treated in three replicates for the future, present, and past $CO_2$ conditions (Engel et al., 2008; Schulz et al., 2008; Riebesell et al., 2007, 2008). Experimental data are available via the data portal Pangaea (doi: 10.1594/PANGAEA.723045 for PeECE II and
doi: 10.1594/PANGAEA.726955 for PeECE III).

Field data of aquatic $CO_2$ concentration, temperature and light were used as direct model inputs (see Appendix C). Measurements of POC, particulate organic nitrogen (PON), and dissolved inorganic nitrogen (DIN) were used for model calibration. Although both the experiments differ in their species composition, environmental conditions and nutrient supply,
same parameter set was used to fit PON, POC, and DIN from PeECE II and III (i.e., a total of 54 series of repeated measures over more than two weeks), a feature indicating the model skills. In addition, the model was validated with another 36 series of biomass and nutrients data from an independent mesocosm experiment (doi: 10.1594/PANGAEA.840852, data not shown). The experimental POC and PON data were redefined for a direct comparison
with model results (see Appendix D), since some contributions (e.g., polysaccharides and transparent exopolymer particles) remain unresolved by our dynamical equations. State variables of our model comprise carbon and nitrogen contents of phytoplankton, $Phy_C$ and $Phy_N$ and DIN, as representative for all nutrients. The dynamics of non-phytoplanktonic components, i.e. detritus and heterotrophs (DH), are distinguished by $DH_C$ and $DH_N$. Thus,
in our study, $POC = Phy_C + DH_C$ and $PON = Phy_N + DH_N$.

The mean cell size in the community, represented as the logarithm of the mean equivalent spherical diameter (ESD), was used as a model parameter. It determines specific ecophysiological features by using allometric relations that are relevant for the computation of subsistence quota, as well as nutrient and carbon uptake rates. Regarding the latter, to
resolve sensitivities to different DIC conditions, we used a relatively accurate description of carbon acquisition as a function of DIC and size. It has been suggested by previous observations and models that ambient DIC concentration increases primary production (e.g. Schluter et al., 2014; Rost et al., 2003; Zondervan et al., 2001; Riebesell et al., 2000; Chen, 1994; Riebesell et al., 1993; Riebesell and Tortell, 2011) and mean cell size in the commu-

nity (Sommer et al., 2015; Eggers et al., 2014; Tortell et al., 2008). While state-of-the-art models such as Artioli et al. (2014) used empirical biomass increase to describe OA effects, we adopted and simplified a biophysically explicit description for carbon uptake from Wirtz (2011), where the efficiency of intracellular DIC transport has been derived as a func-

tion of the mean cell size $\ell = ln(ESD/1\mu m)$ and $CO_2$ concentration. For very large cells, the formulation converges to the surface to volume ratio, which in our notation reads $e^{-\ell}$. In contrast, the dependence of primary production on $CO_2$ vanishes for~~does not apply~~ to picophytoplankton; the rate limitation by sub-optimal carboxylation then reads:

$$f_{CO2} = \Big( \frac{1 - e^{-a_{CO2} \cdot CO_2}}{1 + a^* \cdot e^{(\ell - a_{CO2} \cdot CO_2)}} \Big). \tag{1}$$

The specific carbon absorption coefficient $a_{CO2}$ reflects size-independent features of the DIC acquisition machinery (for instance, the carbon concentration mechanisms (Raven and Beardall, 2003)). The coefficient $a^*$ represents carboxylation depletion.

## 2.2 Uncertainty propagation

We considered that uncertainties were only present in the initial setup of the system; this al-
lowed us to perform a deterministic non-intrusive forward propagation of uncertainty, which neglects the possible coupling between uncertainties and temporal dynamics unlike in intrusive methods (Chantrasmi and Iaccarino, 2012) involving stochastic dynamical equations with time-varying uncertainties (Toral and Colet (2014), M. de Castro et al. in preparation). Forward refers to the fact that unresolved differences among replicates simulated as vari-
ations of the model control factors are propagated through the model to project the overall variability in the system response, (in contrast to backward methods of parameter estimation where the likelihood of input values is conditioned by the prior knowledge of the output distribution (as, for instance, in Larssen et al. (2006)).

Our approach is based on a Monte Carlo method for the propagation of distributions. It is
based on the repeated sampling from the distribution for possible inputs and the evaluation of the model output in each case (JCGM, 2008b). Next, the overall simulated POC variabil-

ity is compared with that in POC experimental data (i.e., the mean trends of the treatment levels as well as the standard deviations are compared, the former for the calculation of the reference run and the latter for the uncertainty propagation). Among the available experimental data, we favored POC over PON and DIN in the uncertainty propagation analysis since it is usually the target variable of OA effects and shows the highest variability. A variability decomposition with more than one dependent variable (equivalent to a multivariate ANOVA design, for instance) is beyond of the scope of the study. The comparison between simulated and experimental variability in POC helps in the identification of the changes in physiological state and community structure that are the main potential contributors to the variability.

We considered model factors, $\phi_i$, with $i = 1, ..., N = 19$, consisting of 14 process parameters and 5 initial conditions for the state variables. Their reference values, $\langle \phi_i \rangle$, were adjusted to yield model solutions reproducing the mean of each treatment level (steps 1 and 2, Tables 1 and 2). To test our first hypothesis, factor variations representing potential uncertainties are introduced as random values distributed around $\langle \phi_i \rangle$ with standard deviation $\triangle \phi_i$. To calculate $\triangle \phi_i$, we first generate $10^4$ simulations, each one with a different factor value, $\phi_i$ (steps 3 and 4). The ensemble of model solutions for each factor and treatment level simulates the potential experimental outcomes, hereafter referred to as "virtual replicates", (see Appendix A). The factor value for each POC trajectory is randomly drawn from a normal distribution around the factor reference value $\langle \phi_i \rangle$ (same distribution is assumed by popular parametric statistical inference tools such as regressions and ANOVA Field et al. (2008)). For every treatment level and at every time step, we calculated the ensemble average of the virtual replicates, $\langle POC_i^{mod} \rangle$, and the standard deviation, $\triangle POC_i^{mod}$. Thus, $\triangle \phi_i$ is the standard deviation of the distribution of factor values such as $\triangle POC_i^{mod}$ does not exceed the standard deviation of the experimental POC data, $\triangle POC^{exp}$, for any mesocosm at any given time (step 5). The effect of variations of $\phi_i$ on the variability (step 6) is given as follows:

$$\varepsilon_i = \frac{\triangle POC_i^{mod}}{\triangle \phi_i}. \tag{2}$$

This ratio expresses the maximum variability a factor can generate, $\triangle POC_i^{mod}$, relative to the associated range of that factor variations, $\triangle \phi_i$, to ensure that $\triangle POC_i^{mod}$ is the closest to $\triangle POC^{exp}$ at any time. In general, $\varepsilon_i$ defines how much of the uncertainty of a dependent variable $Y$ (here $Y =$POC) is explained byand the uncertainty of the input factors $\phi_i$, a proxy of which is known as the sensitivity coefficient $c_i = \frac{\partial Y}{\partial \phi_i}$ in the widespread formula to calculate error propagation (Ellison and Williams, 2012), also known as law of propagation of uncertainty (JCGM, 2008a)

$$(\triangle Y)^2 = \sum_{i=1}^{N} c_i^2 \cdot (\triangle \phi_i)^2.$$

This expression is based on the assumption that changes in $Y$ in response to variations in one factor $\phi_i$ are independent from those owing to changes in another factor $\phi_j$, and that all changes are small (thus cross terms and higher-order derivatives are neglected). Where no reliable mathematical description of the relationship $Y(\phi_i)$ exists (in our case, only an expression for the rate equation dPOC/dt is known (see Table 1) but not its analytical solution, i.e., POC), $c_i$ can be evaluated experimentally (Ellison and Williams, 2012; JCGM, 2008a). As mentioned in the Introduction and Appendix A, such high-dimensional multi-factorial measurements are costly in mesocosm experiments. Therefore, we obtained equivalent information by numerically calculating $\varepsilon_i$. Such approximations to sensitivity coefficients calculated by our Monte Carlo method of uncertainty propagation correspond to taking all higher-order terms in the Taylor series expansion into account since no linearization is required (see Sections 5.10 and 5.11 and Annex B in JCGM (2008b)). A straightforward extension including the cross terms showing synergistic uncertainties effects, as in an experimental multi-way ANOVA design, requires the assumption of joint distributions for the uncertainty of factors and the calculation of covariance matrices, a considerable effort that is beyond of the scope of this paper.

Hereafter, the standard deviation of any given factor, i.e., factor uncertainty, will be given as percentage of the reference values and will be called $\triangle \Phi_i$. The actual factor range is

given as $\triangle\phi_i = \frac{\triangle\Phi_i \cdot \phi_i}{100}$. Strong irregularities in the standard deviations of experimental POC data (for instance, small $\triangle POC^{exp}$ at day 8 in Fig.2p), translates to remarkably enhanced or reduced sensitivity coefficients if the model-data comparison would be performed at a daily basis. Therefore, we considered the temporal mean of the standard deviation per
phase, i.e., prebloom, bloom, and postbloom. We inferred phases for PeECE II from Engel et al. (2008) and for PeECE III from Schulz et al. (2008) and Tanaka et al. (2008).

    To numerically calculate the ensemble of $10^4$ POC trajectories per factor (i.e., the virtual replicates, see Fig. 8), we applied the Heun integration method with a time step of $4 \cdot 10^{-4}$, (about 35 s of experimental time). The number of simulated POC time-series is chosen
such as a higher number of model realizations, i.e., a higher number of virtual replicates, will produce the same results (see Adaptive Monte Carlo procedure, Section 7.9. in JCGM (2008b)). We dismissed the negative values that randomly appeared when drawing $10^4$ values from the normal distribution of factor values; this reduction in the number of trajectories did not affect the results.

Environmental data showed low variability among same treatment replicates, (see Fig. 9), suggesting a non-direct relation between variations in environmental factors among replicates and the observed biomass variability. Therefore, we focused in uncertainties in ecophysiology and community composition and used environmental data as forcing. Perturbations of the similarity among replicates produced by strong changes in environmental
conditions (storms, dysfunctional devices, etc.) or by errors in manipulation or sampling procedures are not the scope of this work. After a few decades, the current state-of-the-art of experimental techniques for running plankton mesocosms is advanced. We believe such differences are of low impact or well-understood in terms of their consequences for final outcomes (Riebesell et al., 2010; Cornwall and Hurd, 2015).

Notably, our analysis suggested sufficient (but not necessary (Brennan, 2012)) causes of uncertainties in mesocosm experiments. Variations in model characterization including structural variability (Adamson and Morozov, 2014; Fussmann and Blasius, 2005) or uncertainties in model parametrization (Kennedy and O'Hagan, 2001) or comparisons to different uncertainty propagation methods (M. de Castro et al, in preparation) require further exten-

sive analyses beyond the scope of this study. However, we performed a series of preceding model analyses (including uncertainty propagation) by using slightly different model formulations (data not shown). From these preceding analyses, we found that different model formulations can lead to quantitatively different confidence intervals, but leave the final re-
sults qualitatively unchanged.

## 3 Results

### 3.1 $CO_2$ effect on POC dynamics

Our model reproduces the means of PON, POC and DIN experimental data per treatment level, i.e., for the future, present, and past $CO_2$ conditions, in two independent PeECE
experiments (Figures 2 and 3). For PeECE II, PON is moderately overestimated and post-bloom POC is slightly underestimated. Nonetheless, the model represents the experimental data with similar precision than the means of experimental replicates (see Appendix E). The means of the same treatment replicates and their associated standard deviations are typically used to represent experimental data (see Figure 1b in Engel et al. (2008) for PeECE
II or Figure 8a in Schulz et al. (2008) for PeECE III). The means are in the foundations of the statistical inference tools that did not detect acidification responses for PeECE II and III. However, with our mechanistic model-based analysis, phytoplankton growth in the future $CO_2$ conditions showed an earlier and elevated bloom with respect to past the $CO_2$ conditions. The future and past reference trajectories limit the range of the $CO_2$ enrichment effect,
as shown by the dark gray area in Figure 4. POC variability owing to variations in model factors simulating experimental uncertainties is plotted as the light gray area in the figure. Our results suggest that avoiding high POC standard deviations that potentially mask OA effects in experimental data requires the reduction of the factor variations triggering variability during the bloom.

## 3.2 CO$_2$ effect on uncertainty propagation

The estimation of the tolerance thresholds of the dynamics to uncertainty propagation for the two test-case experiments, per acidification levels and per factor uncertainty, are listed in Table 3. We investigated the potential interaction of the treatment and the uncertainties effects on the tolerance by a linear mixed-effects model with $\phi_i$ as random factor (R Core Team, 2016). The synergistic effect between the factor uncertainty and the treatment levels was found to be non significant (F=2.9 with p=0.06). Therefore, the thresholds do not appear to statistically depend on the treatment level, even when the standard deviation of the measured POC data, $\triangle POC^{exp}$, for the future and past acidification conditions were, on average, about 70% larger than the standard deviation of the present conditions (POC experimental data in Figs. 2 and 3 are more spread in the future and past concentrations than in the present concentration). Despite the low statistical power of this test (only data from two independent samples, the two PeECE experiments, were available), we still considered the potential lack of CO$_2$ effect on the uncertainty propagation as sufficient justification to simplify further analysis on variability decomposition by averaging the thresholds and the sensitivity coefficients over treatment levels (see last column in Table 3 and Fig. 7).

## 3.3 Variability decomposition

Our method allows decomposition of POC variability in factor-specific components $\triangle POC_i^{mod}$. The effect of factor variations simulating experimental differences among replicates is classified depending on its nature, intensity and timing (Figures 5 and 7).

POC variability during the prebloom phase can be explained mainly by the differences of factors related to subsistence quota, i.e., $Q_{subs}^*$ and $\alpha_Q$, in both PeECE II and III experiments (left column in Fig. 5). This suggests that the differences in subsistence quota first intensify the divergence of POC trajectories and then weaken few days later because of the system dynamics. These subsistence parameters only need to vary about 6% and 8% among replicates (see Table 3) to maximize their contribution to the $\triangle POC^{exp}$; thus, their sensitivity coefficients are the highest (see Fig.7).

Differences in the initial nutrient concentration, DIN(0), mean cell size, $\ell$, and phytoplankton biomass loss coefficient, $L^*$, generate the modeled variability mainly during the bloom (with just about 20% differences among replicates, see Table 3 and second column in Fig. 5) showing high values of the sensitivity coefficient (gray highlight in Fig. 7).

Amplified variability in the postbloom phase (third column in Fig. 5) can be attributed to the uncertainties in the reference temperature $T_{ref}$ for the Arrhenius equation, Eq. (B2), in sinking loss or export flux, s, and in remineralization and excretion, $r^*$. The sensitivity coefficient of $T_{ref}$ is high, with just about 12% variation. Therefore, even if differences in ambient temperature among replicates of the same sample are negligible (see the low standard deviations in the temperature, Fig. 9), differences in the metabolic dependence on that ambient temperature seems to be relevant in the decay phase. Interestingly, variations among replicates in the physiological dependence on other environmental factors do not show the same relevance (the sensitivity coefficient $\varepsilon_i$ is low for carbon acquisition $a_{CO2}$ and light absorption $a_{PAR}$). Generating high divergence during the postbloom requires a strong perturbation of parameters for the description of the non-phytoplanktonic biomass (about 81% of the reference value for sinking and 96% for remineralization and excretion, see Table 3), which translates to a relatively low sensitivity coefficient.

Perturbations of the initial detritus concentration, $DH_C(0)$ and $DH_N(0)$ have no impact on the dynamics provided that they remain within reasonable ranges ($\triangle\Phi_i < 100$). In fact, more than tenfold difference among replicates in such non-relevant factors were necessary to achieve a perceptible variability $\triangle POC_i^{mod}$.

POC variability throughout the bloom phases (right column in Fig. 5) can be attributed to the varying carbon and nitrogen initial conditions, $Phy_C$ and $Phy_N$, nutrient uptake-related factors, $V_{max}^*$, $\alpha_V$, and Aff, and protein allocation for photosynthetic machinery, $f_p$. About the latter, high standard deviations of the tolerance (see Table 3) suggest non-conclusive results.

## 4 Discussion

We used the uncertainty quantification method to decompose POC variability by using a low-complexity model that describes the major features of phytoplankton growth dynamics. The model fits the mean of mesocosm experimental PeECE II and III data with high accuracy for all $CO_2$ treatment levels. We confirmed the working hypotheses (Figs. 5,7); in particular, we showed that small differences in initial nutrient concentration, mean cell size and phytoplankton biomass losses are sufficient to generate the experimentally observed bloom variability $\triangle POC^{exp}$ that potentially mask acidification effects, as discussed in the following subsections.

The results of our analyses are conditioned by the dynamical model equations imposed. Deliberately, the model's complexity is kept low, mainly to limit the generation of structural errors with respect to model design. At the same time, the level of complexity resolved by the model suffices to explain POC measurements of two independent mesocosm experiments with identical parameter values (see Table 2), which highlights model skill. The used equations comply with theories of phytoplankton growth (e.g. Droop, 1973; Aksnes and Egge, 1991; Pahlow, 2005; Edwards et al., 2012; Litchman et al., 2007; Wirtz, 2011). The uncertainty propagation employed here can be applied to any model. As long as the model features a similar structural complexity and is also able to reproduce POC with sufficient accuracy, we expect similar qualitative findings with respect to the factors ($\Phi_i$) and similar identification of the major contributors to the variability. However, we would not expect other models to reveal exactly similar values in the ratio $\epsilon_i$, which would likely depend on the equations used to resolve some of the ecophysiological details.

### 4.1 Nutrient concentration

Differences among replicates in the initial nutrient concentration substantially contribute to POC variability, a sensitivity that is, interestingly, not well expressed when varying the initial cellular carbon or nitrogen content of the algae, $Phy_C(0)$ and $Phy_N(0)$. The relevance of accuracy for the initial nutrient concentration in replicated mesocosms has already pointed in

Riebesell et al. (2008). Under a constant growth rate, DIN(0) determines the timing of nutrient depletion; therefore, differences in the initial nutrient concentration might also translate into temporal variations in the succession of species. We showed that such dependence is noted even in more general dynamics, and that our method can also estimate the varia-
tional range for differences in the initial DIN concentration for experiments with low number of replicates. The standard deviation of DIN(0) in the experimental setup for PeECE III was 50% of the mean, which is significantly above our tolerance threshold (see Table 3 for initial DIN concentration). Following Riebesell et al. (2007), we considered day 2 as the initial condition, when the measured DIN was $14 \pm 2$ $\mu$mol-C L$^{-1}$, as shown in Table 1. Since 2
$\mu$mol-C L$^{-1}$ is approximately the 14% of 14 $\mu$mol-C L$^{-1}$, the variability of replicates at day 2 was about 14%. Therefore, experimental differences in the initial nutrient concentration were similar to the tolerance threshold for the initial DIN established to avoid high variability ($(20 \pm 6)$% in Table 3), which represents an explanation for the high divergence observed in POC measurements.

For PeECE II, experimentally measured DIN concentration at day 0 was $10.7 \pm 0.8$ $\mu$mol-C L$^{-1}$, suggesting a 7.5% difference among replicates, which was below our projected tolerance level (7.5 is out of the range $[14, 26]$). The same was noted for day 2, with DIN concentration equal to $8 \pm 0.5$ $\mu$mol-C L$^{-1}$ (Table 1). Our approach showed that differences in initial nutrient concentration in PeECE II were not sufficiently high to trigger the experimentally
observed POC variability. Incidentally, phosphate re-addition on day 8 of the experiment established new initial nutrient concentration for the subsequent days. When the dynamics in one replicate significantly diverges from the mean dynamics of the treatment, even if the re-addition occurs at the same time and at the same concentration in all the replicates, the mesocosm with the outlier trajectory will not respond as the others do, and with the ad-
dition of a new nutrient condition, the divergence might be further amplified. In the case, nutrient re-addition has the same impact on the systems as variations in the initial conditions of nutrient concentration. Also for PeECE II, variability in POC is about 30% higher than that for PON, as shown in Fig. 2. We attribute the temporal decoupling between C and N dynamics to the break of symmetry among replicates by the nutrient re-addition owing

to the strong sensitivity of the system to initial nutrient concentrations and a concomitant change in subsistence N:C quota, which is a sensitive parameter, especially during the pre-bloom phase (Fig. 5 and Fig. 7). Increase of POC:PON ratios under nitrogen deficiency has been observed frequently during experimental studies (e.g. Antia et al., 1963; Biddanda

and Benner, 1997) and has been attributed to preferential PON degradation and to intracellular decrease of the N:C ratio (Schartau et al., 2007). Hence, we confirmed that nutrient re-addition during the course of the experiments results in a significant disturbance, as has been previously reported Riebesell et al. (2008), although a complete analysis would require a model that explicitly accounts for other nutrients.

**4.2 Mean cell size as a proxy for community structure**

We found a limited tolerance to variations in the mean cell size of the community, $\ell$, which has a threshold of about $22\%$ variation (see Table 3). If we consider the averaged mean cell size of PeECE II, $\langle\ell\rangle = 1.6$, and III, $\langle\ell\rangle = 1.8$, from Table 2, we obtain $\langle\ell\rangle = 1.7$. Then the absolute standard deviation is $\triangle\ell = 22 \cdot \frac{1.7}{100} \sim 0.4$. Therefore, our methodology shows that

variations within the range limited by $\langle\ell\rangle \pm \triangle\ell$, that is $[1.3, 2.1]$, are sufficient to reproduce the observed experimental POC variability during the bloom. Since $\ell$ is in the log-scale, the corresponding ESD increment is within the variational range $\langle\mathrm{ESD}\rangle \pm \triangle\mathrm{ESD}$, that is, $[3.7, 8.1]\mu m$ (or $[25, 285]\mu m^3$ in volume). These values are easily reached in the course of species succession. This supports studies showing that community composition outweighs

ocean acidification (Eggers et al., 2014; Kroeker et al., 2013; Kim et al., 2006).

**4.3 Phytoplankton loss**

Another major contributor to POC variability during the bloom phase is phytoplankton biomass loss, L$^*$. With a standard deviation of about $20\%$ (Table 3), uncertainties in L$^*$ generate variability larger than the model response to OA, in particular at the end of the growth phase

and the beginning of the decay phase. Unresolved details in phytoplankton loss rate include, among others, replicate differences in cell aggregation or damage by collisions, mortality

by virus, parasites, and morphologic malformations, or grazing by non-filtered mixotrotophs or micro-zooplankton.

### 4.4 Inference from summary statistics on mesocosm data with low number of replicates

To test the hypotheses outlined in the Introduction entails two important aspects. First, heuristic exploration of variability would require experiments designed to quantify the sensitivity of mesocosms to variations in potentially relevant factors that specify uncertainties in environmental conditions, cell physiology, and community structure. However, this would require high-dimensional multi-factorial setups (see Appendix A), which would be difficult to handle, if at all, even for low number of replicates. Second, standard statistical inference tools might come to their limitations in estimating treatment effects. Repeated measures of relevant ecophysiological data (e.g., POC) are collected from mesocosm experiments that span a few weeks. If the differences among treatment levels are smaller than those among replicates of the same treatment level, post-processing statistical analyses might conclude that there are no detectable effects (Field et al., 2008).

In many cases, the mean and the variance of the sample are taken as a fair statistical representation of the effect of the treatment level and its variability. However, summary statistics such as the mean and the variance might fail to describe distributions that do not cluster around a central value, i.e, when the data are not normally distributed in the sample. This is because a feature of normally distributed ensembles is that the mean represents the most typical value and deviations from that main trend (caused by unresolved factors not directly related to the treatment) might cancel out in the calculation of the ensemble average. Actually, this cancellation is the reason for using replicates (Ruxton and Colegrave, 2006), but many circumstances can remarkably lower the likelihood for cancellation, for instance: (i) effects that are sensitive to initial conditions (thus, small initial differences in the replicates of a given sample might become amplified and produce departures that enlarge over the course of the experiment); (ii) non-symmetrically distributed initial conditions in the sample (that might lead to non-symmetrical distribution of the results); and (iii) a low

number of replicates, i.e., a sample size not adapted to the intensity of the treatment effect, the sensitivity of all effects to initial conditions, and the intended accuracy of the experiment. Each incident decreases the statistical power and therefore misleading conclusions might be inferred (Miller, 1988; Cohen, 1988; Peterman, 1990; Cottingham et al., 2005).

## 4.5 Consequences for the design of mesocosm experiments

In our simulations, the $CO_2$ level affected the intensity and timing of the bloom (Figure 4). Thus, the slope of the growth phase can be regarded as a suitable target variable to detect OA effects. Moreover, our model analysis revealed a low signal-to-noise ratio. The ability to distinguish the treatment effect from noise depends on the experimental design, the strength of the treatment, and the variability that it is not explained by the treatment. When the signal-to-noise ratio is as low as it is shown by our simulations, a large experimental sample size is needed to avoid to incur in a Type II error (Field et al., 2008). In particular, we can assume a two sample two sided balanced t-test with two treatment levels as in Figure 4. The maximum difference between means equal to approximately $5\,\mu$mol-C L$^{-1}$ (see for instance PeECE III at day 10) and the variability $\triangle$POC$^{\text{mod}}$ approximately equal to $4\,\mu$mol-C L$^{-1}$. If we aim for a statistical power of 0.8, .i.e., a $80\%$ chance of observing a statistically significant result with that experimental design, the required number of replicates per treatment would be 11 (R Core Team, 2016), which is unpractical in mesocosm experiments. With n=3 replicates, the chance declines to only $20\%$.

We provided an estimation for the uncertainty thresholds that can be used for improving future sampling strategies with low number of replicates, i.e. n=3. Tolerances shown in Table 3 can be used to quantify how much replicates similarity can be compromised before the variability of the outcomes outweighs potential acidification effects. Some tolerances indicate maximal variations in observable quantities, such as nutrient concentration and community composition. These model results suggest that a better control of such dissimilarities among replicates can help maintain the variability below the range of the acidification effect, especially during the bloom.

Strategies to reduce $\triangle POC^{mod}$ should similarly apply to lower $\triangle POC^{exp}$. For example, model results turned out to be very sensitive to variations in mean logarithmic cell size. Variations of this factor during the initial filling of the mesocosms may already generate divergent responses in POC so that a potential $CO_2$ signal becomes difficult to detect, if at all. To determine spectra of cell sizes (or mean of logarithmic cell size) of the initial plankton community prior to $CO_2$ perturbation would be a possibility to countervail this difficulty. The decision of which mesocosm to select for which kind (i.e. intensity) of perturbation may then be adjusted according to similarities in initial plankton community structure. For example, we may consider some number of available mesocosms that should become subject to two different $CO_2$ perturbations. We may first select two mesocosms that reveal the greatest similarity with respect to their initial size spectra and assign them to the two different $CO_2$ treatments. Likewise, from the remaining mesocosms we again chose those two mesocosms that show closest similarity between their size spectra. Those two are chosen to become subject to the two different $CO_2$ perturbations. The selection procedure could be repeated until all mesocosms have been assigned to either of the two $CO_2$ treatments. Thus, mesocosms with similar initial conditions are assured to become subjected to different $CO_2$ perturbations. This reduces a mixture of random effects due to variations in experiment initialization and $CO_2$ effect and it will likely facilitate data analysis in (in experimental setups with low number of replicates, where sample randomization (Ruxton and Colegrave, 2006) might not be effective, see Section 4.4). Mesocosms may then be first analyzed pairwise (similar initial setup) with respect to differences in $CO_2$ response.

In addition, our analysis results help interpreting non-conclusive results and provide plausible explanations for the negative results for the detection of potential acidification effects (Paul et al., 2015; Schulz et al., 2008; Engel et al., 2008; Kim et al., 2006; Engel et al., 2005). Thus, our study also suggests the limitation of the statistical inference tools commonly used to assess the statistical significance of effect detectability.

Finally, we found the same main contributors to POC variability for all the treatment levels, even if experimental variability is about 70% higher in the mesocosms where the carbon chemistry was manipulated. In particular, the heterogeneity of variance measured in future

levels is larger than under the other acidification conditions (see fluctuations of the standard deviations of $CO_2$ concentrations, Fig. 9). These differences in biomass variability among treatment levels are not explained by uncertainties in our model factors. They might have been originated by the irregularities in the $CO_2$ aeration (Riebesell et al., 2008; Cornwall and Hurd, 2015), however, further analyses need to be conducted to determine potential sources of *differences* in variability.

## 5 Conclusions

Our model projections indicated that phytoplankton responses to OA were mainly expected to occur during the bloom phase, presenting a higher and earlier bloom under acidification conditions. Moreover, we found that amplified POC variability during the bloom that potentially reduces the low signal-to-noise ratio can be explained by small variations in the initial DIN concentration, mean cell size, and phytoplankton loss rate.

The results of the model-based analysis can be used for refinements of experimental design and sampling strategies. We identified specific ecophysiologial factors that need to be confined in order to ensure that acidification responses do not become masked by variability in POC.

With our approach we reverse the question of how experimental data can constrain model parameter estimates and instead determine the range of variability in experimental data that can be explained by modeling with variational ranges bounding uncertainties of specific control factors. We tested the hypothesis whether small differences among replicates have the potential to generate higher variability in biomass time-series than the response that can be attributed to the effect of $CO_2$. Therefore, we conclude that modeling studies that integrate data from acidification experiments should resolve physiological regulation capacities at cellular and community levels. In fact, modeling the propagation of uncertainties revealed cell size to be a major contributor to phytoplankton biomass variability. This suggests themotivates to use of adaptive size-trait based dynamics since such approaches allow for the resolution of ecophysiologial trait shifts in non-stationary scenarios (Wirtz, 2013). The role

of intracellular protein allocation can also be clarified by using a trait-based approach, since our results about the impact of its variations were non-conclusive.

In this study, we established a foundation for further model-based analysis for uncertainty propagation that can be generalized to any kind of experiments in biogeosciences. Extensions comprising time-varying uncertainties by introducing a new random value for parameters at every time step or including covariance matrices, showing the simultaneous interaction of variations in two factors, can be straightforward implemented (M. de Castro et al. in preparation). Finally, we believe that an explicit description of uncertainty quantification is essential for the interpretation and generalization of experimental results.

## Appendix A: Model representation of replicates

Heuristic exploration of the potential origins of the observed variability uses statistical inference tools such as a multi-way repeated measures ANOVA exploring which independent factors are contributing the most to the standard deviations. Such approaches have the advantage of accounting for interacting effects between combinations of factors (and not only for the synergistic effects of each factor and acidification, as in our model-based approach, see Sec. 3). However, the realization through and experimental setup would require a high-dimensional multi-factorial experiment extremely difficult to perform (Fig. 8). For three acidification levels, the minimum number of factor levels (i.e., high and low), minimum number of sample units (i.e., duplicates), and the same number of factors we analyze here, (i.e., N=19), the total number of mesocosms would be $3 \times 2 \times 2 \times 19 = 228$. The possibility of simulating a high number of replicates is one of the unique strengths of modeling. For each factor, we simulate possible realizations of the same acidification level with slight variations of the factor reference value (simulating differences in physiological states and community structure). We generated model solutions for $10^4$ normally distributed factor values i.e., in total, 3 acidification levels x 19 factors x $10^4$ virtual replicates for PeECE II and III experiments. Examples of 50 virtual replicates with uncertainty in initial nutrient concentration are shown in Figure 8 and examples of 10 virtual replicates with uncertainty in phytoplankton

biomass losses are shown in Figure 1, both numerically calculated for low $CO_2$ conditions in PeECE III.

## Appendix B: Definition of relative growth rate

Relative growth rate $\mu$ is calculated from the primary production rate by subtracting respiration and mortality losses as follows: $\mu = P - R - L$.

### Primary production

Primary production rate reflects the limiting effects of light, dissolved inorganic carbon (DIC), temperature and nutrient internal quota as follows:

$$P = P_{max} \cdot f_{PAR} \cdot f_{CO2} \cdot f_T \cdot f_Q \cdot f_p. \tag{B1}$$

$P_{max}$ is the maximum primary production rate, (Table 2). Specific light limitation $f_{PAR}$ depends on light and $CO_2$. For the attenuation coefficient $a_z$, we consider that in coastal regions light intensity is typically reduced to $1\%$ of its surface value at 5 m (Denman and Gargett, 1983) and we obtained $a_z = 0.75 m^{-1}$. Next, PAR experienced by cells at mixed layer depth (MLD$= 4.5\,m$, Engel et al. (2008)), was calculated from the level of radiation at the water surface, $PAR_0$ (see Appendix C), following an exponential decay described by the Lambert-Beer law

$$PAR = PAR_0 \int\limits_0^{MLD} e^{-a_z \cdot z} dz.$$

The relationship between photosynthesis and irradiance can be formulated by referring to a cumulative one-hit Poisson distribution (Ley and Mauzerall, 1982; Dubinsky et al., 1986). With the temperature and carbon acquisition dependence, it yields

$$f_{PAR} = \left(1 - e^{-\dfrac{a_{PAR} \cdot PAR}{P_{max} \cdot f_{CO2} \cdot f_T}}\right),$$

where $a_{PAR}$ is the effective absorption related to the chloroplast cross-section and saturation response time for receptors (Geider et al. (1998a),Wirtz and Pahlow (2010)); the carbon acquisition term $f_{CO2}$ is described in Section 2.1, Eq. (1).

$f_T$ is the temperature dependence. We considered that all metabolic rates depend on pro-
tein folding that increases with rising temperature following the Arrhenius equation (Scalley and Baker, 1997) as described in Geider et al. (1998b) or Schartau et al. (2007)

$$f_T = e^{-E_a \cdot \left( \frac{1}{T} - \frac{1}{T_{ref}} \right)}, \tag{B2}$$

with activation energy $E_a = \frac{T_{ref}^2}{10} \cdot log(Q_{10})$ as in Wirtz (2013), where we used $Q_{10} = 1.88$ for phytoplankton (Eppley, 1972; Brush et al., 2002) and $T_{ref}$ was the mean measured temper-
ature (see Appendix C).

The allometric factor $\alpha_Q$ quantifies the scaling relation of subsistence quota and cell size. We used Droop dependency on nutrient N:C ratio (Droop, 1973), which has been recently mechanistically derived (Wirtz and Pahlow, 2010; Pahlow and Oschlies, 2013)

$$f_Q = \left( 1 - \frac{Q_{subs}}{Q} \right)$$

where $Q = \frac{Phy_N}{Phy_C}$. Its lower reference, the subsistence quota $Q_{subs} = Q_{subs}^* \cdot e^{-\alpha_Q \cdot \ell}$, is con-
sidered size-dependent and reflects a lower protein demand for uptake mechanisms in large cells (Litchman et al., 2007).

The last term in Eq. (B1) accounts for an energy allocation trade-off in phytoplankton cells: protein allocation for photosynthetic compounds such as RuBisCo and pigments, $f_p$,
versus allocation for nutrient uptake, $f_v$, expressed by $f_p + f_v = 1$ (Wirtz and Pahlow, 2010; Pahlow and Oschlies, 2013). We simplified the detailed partition models by setting the trait fractions as constant.

**Respiratory cost and nutrient uptake rates**

Efforts related to nutrient uptake $V$ are represented by a respiration term. Other expenses such as biosynthetic costs are neglected (Pahlow, 2005). Respiration rate is then calculated as $R = \zeta \cdot V$ where $\zeta$ expresses the specific respiratory cost of nitrogen assimilation (Raven,
1980; Aksnes and Egge, 1991; Pahlow, 2005). For simplicity, our model merges the set of potentially limiting nutrients (e.g. P, Si and N) to a single resource only, that is DIN. We follow Aksnes and Egge (1991) as described in Pahlow (2005) for the maximum uptake rate

$$V_{max} = \frac{1}{\dfrac{1}{V_{max}^* \cdot f_T} + \dfrac{1}{Aff \cdot DIN}},$$

comprising the maximum uptake coefficient $V_{max}^*$ and nutrient affinity Aff. In addition to
a temperature dependence of nutrient uptake as reported by Schartau et al. (2007), we assumed that respiratory costs decrease with increasing cell size (Edwards et al., 2012), which leads to an allometric scaling in nutrient uptake(Wirtz, 2013) with exponent $\alpha_V$. We also accounted for the static proteins allocation trade-offs between photosynthetic machinery, $f_p$, and nutrients uptake, $f_v = 1 - f_p$. Thus, the nutrient uptake term yields

$$V = (1 - f_p) \cdot V_{max} \cdot e^{-\alpha_V \cdot \ell}.$$

**Loss rates**

To describe the loss rate of phytoplankton biomass, we used a density-dependent term

$$L = L^* \cdot (Phy_C + DH_C).$$

The resulting matter flux increases the biomass of detritus and heterotrophs (DH), and a
fraction of it becomes a part of the remineralizable pool. A temperature-dependent remineralization term (Schartau et al., 2007)

$$r = r^* \cdot f_T$$

describes any kind of DIN production, such as hydrolysis and remineralization of organic matter, excretion of ammonia directly by zooplankton, and rapid remineralization of fecal pellets produced also by the zooplankton. The other fraction of the non-phytoplanktonic biomass is removed by settling with a rate related to the sinking coefficient, s, shown in Tables 1 and 2. Our model was calibrated with experimental data from enclosed mesocosms where aquarium pumps ensured mixing. Therefore, we assumed that sufficiently wealthy organisms could achieve neutral buoyancy (Boyd and Gradmann, 2002), and thus sinking might not have directly affected the phytoplankton biomass.

## Appendix C: Forcings

We used measured aquatic $CO_2$ and temperature per mesocosm and ambient PAR, as model inputs (see Fig. 9). For the two PeECE experiments, the photon flux density was measured by the Geophysical Institute of the University of Bergen. To calculate the surface radiation inside the mesocosms, $PAR_0$, we followed (Schulz et al., 2008) and considered that $80\%$ of incident PAR passed through the gas tight tents, of which up to $15\%$ penetrated to approximately 2.5 m depth, the center of the mixed surface layer in PeECE III. The daily carbon dioxide data were interpolated and PAR signal was filtered by singular spectrum analysis to avoid sudden changes that could be detrimental for the performance of the numerical calculation, since the Heun method requires differentiable functions.

## Appendix D: Definition of POC

The applied model equations attribute phytoplankton, detritus, and herbivorous heterotrophs to particulate organic matter. Measurements of particulate organic carbon also include some fractions of large bacterioplankton, carnivorous zooplankton, as well as extracellular gel particles such as transparent exopolymer particles. These additional organic contributions to POC measurements are not explicitly resolved in our model. Therefore, for comparisons between simulation results and observations, we redefine the raw data from

PANGAEA, named POC' hereafter (dots in Figs. 2, 3 and 5 represent the already modified POC data). We used data of transparent exopolymer particles, TEP, from Egge et al. (2009) for PeECE III , such as here POC = POC' - TEP. For PeECE II, TEP data were not available. We used POC = POC'- POC", where POC" is the difference between particle abundance, PA, of the Coulter Counter measurements and the Flow Cytometry data in Engel et al. (2008):

$$POC" = \beta \cdot (PA_{\text{Coulter Counter}} - PA_{\text{Flow Cytometry}}).$$

The scaling parameter $\beta$=0.000065 $\mu$mol-C$^{-1}$ L was tuned to provide reductions between 40 and 50% from total POC, in agreement with the adjustments of PeECE III.

## Appendix E: Residuals of the model-data fit

For the model-data fit shown in Figs. 2 and 3, we calculated the cumulative residuals E and M (Table 4) with respect to the mean of experimental replicates per treatment, time and mesocosm. For experimental data, residuals E were calculated as follows:

$$E = \sum_{\text{treat,rep,day}} |Y^{\text{exp}}_{\text{treat,rep,day}} - \langle Y^{\text{exp}}_{\text{treat,day}} \rangle| / \eta$$

and for model results, residuals M were calculated as follows:

$$M = \sum_{\text{treat,rep,day}} |Y^{\text{mod}}_{\text{treat,rep,day}} - \langle Y^{\text{exp}}_{\text{treat,day}} \rangle| / \eta$$

with $\eta = 9$ being the total number of mesocosms. High residuals entail high deviation from the trend. In the case of E, this is the deviation from the mean of the treatment (typically used in statistical inference tools), and in the case of M, the deviation from the model reference run. When both E and M values are comparable, we can infer that the quality of both representations is similar (see Table 4). Thus, conclusions inferred from both approaches are based on equally valid assumptions.

*Author contributions.* K.W., M.S., and M.M.C. developed the model code; M.M.C performed the simulations and prepared the manuscript, which was revised by K.W. and M.S.

*Acknowledgements.* We thank Sabine Mathesius for the PAR and temperature data for both the PeECE II and III experiments and Kaela Slavik for the English edition of the preliminary version of the manuscript. We acknowledge our two anonymous reviewers for their helpful comments and suggestions. This work is a contribution to the National German project Biological Impacts of Ocean Acidification (BIOACID) and it is also supported by the Helmholtz society via the program PACES.

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

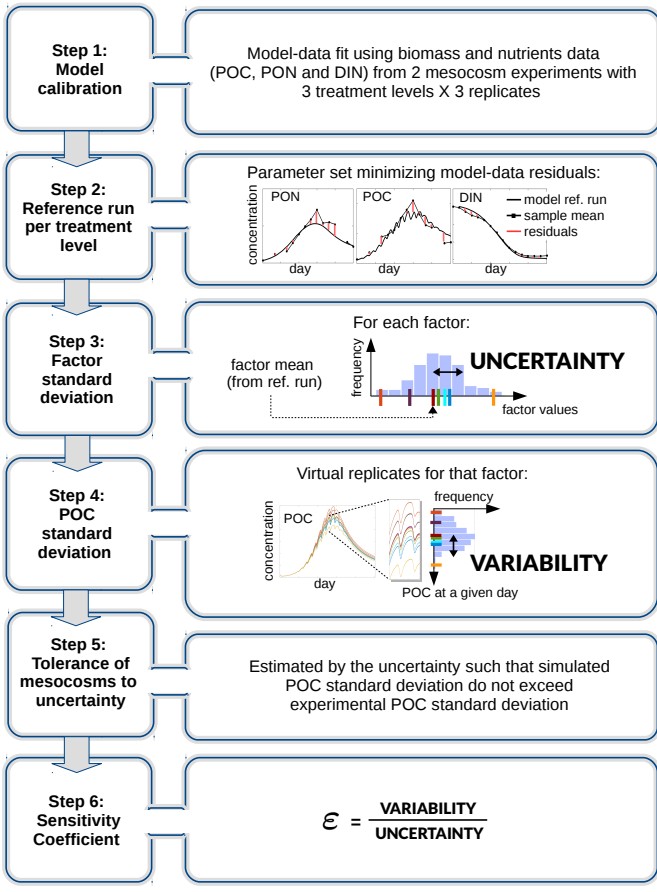

**Figure 1.** Variability decomposition method based on uncertainty propagation (summary of the basic principles given in Sections 5.1.1 and 5.6.2. and Annex B in JCGM (2008b)).

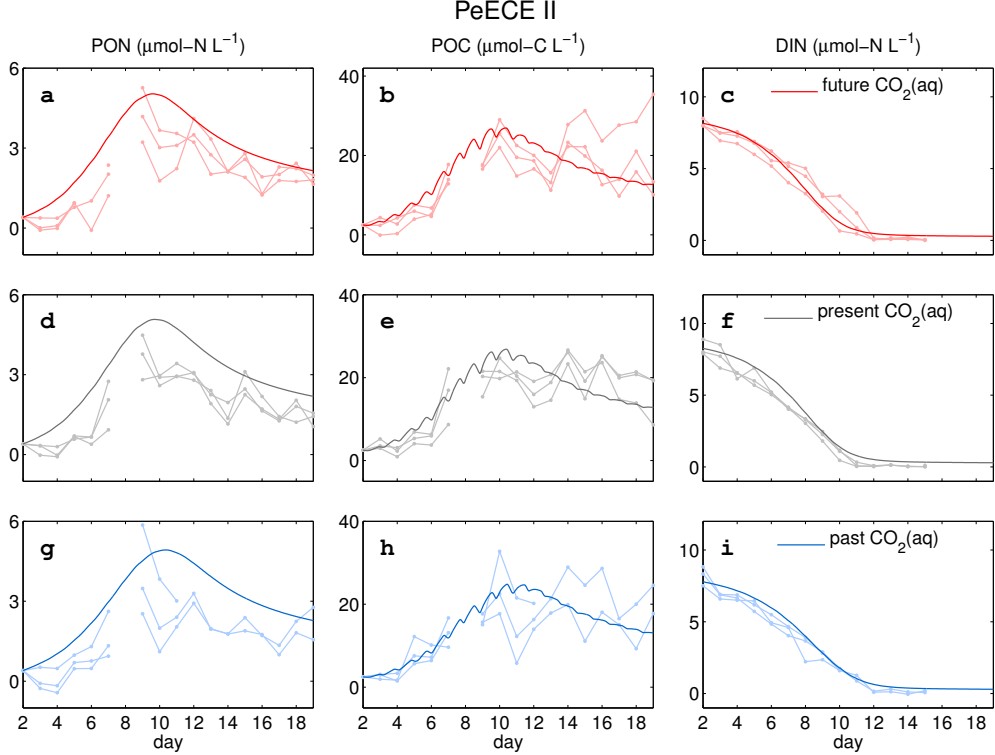

**Figure 2.** Solid lines show reference runs for POC, PON and DIN simulating the mean of the replicates per treatment level, with different colors for the three experimental $CO_2$ setups. Dots are replicated data from the Pelagic Enrichment $CO_2$ Experiment (PeECE II) for newly produced POC and PON, i.e. starting values at day 2 were subtracted from subsequent measurements as in Riebesell et al. (2007).

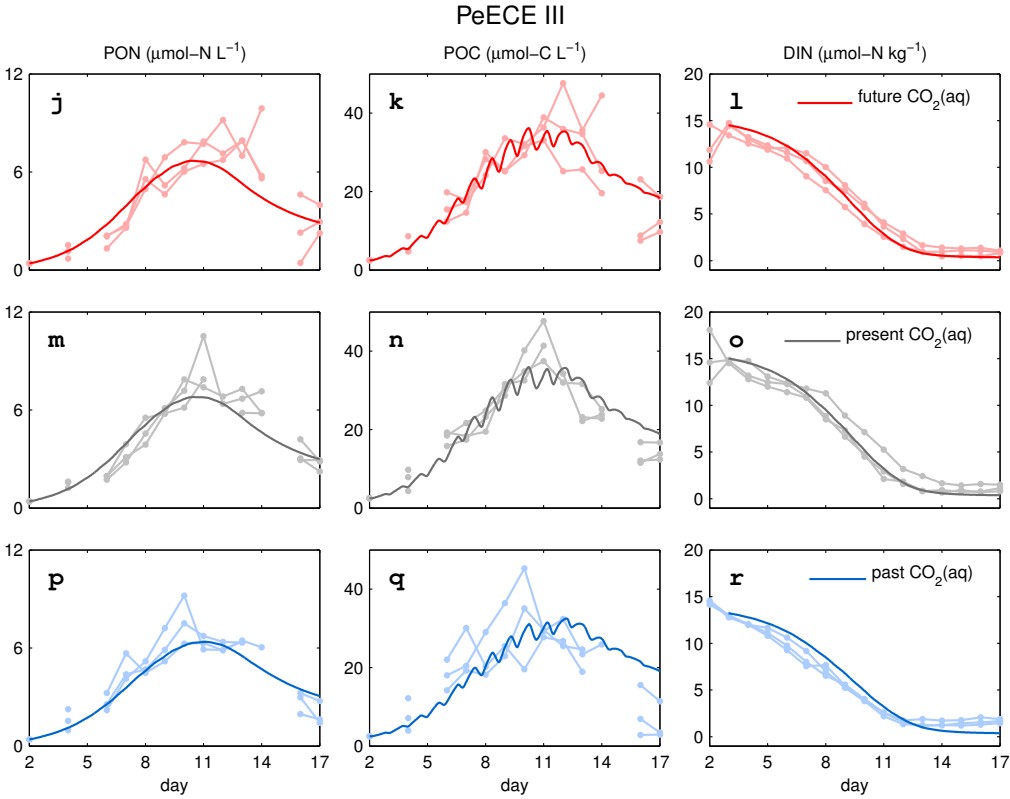

**Figure 3.** As in Fig. 2 for PeECE III.

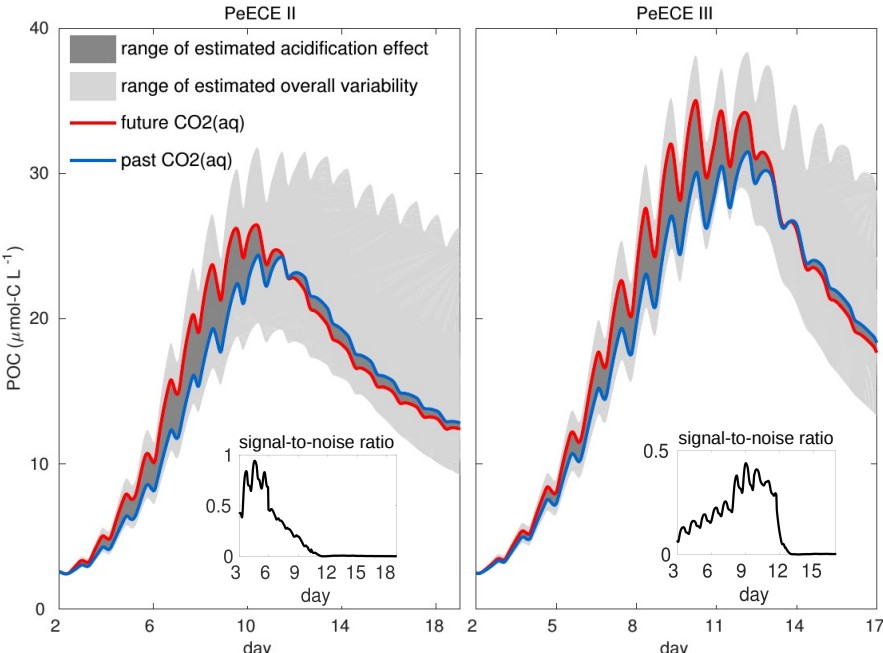

**Figure 4.** Reference simulations of POC for high $CO_2$ (red) and low $CO_2$ (blue) conditions bind the range of acidification effect (dark gray) according to our model projections. Light gray area shows the limits of the overall simulated POC variability, $\triangle POC^{mod}$. Inlay graph display the signal-to-noise ration (black solid lines), i.e., the ratio between the variance of the acidification effect and the variance of the overall variability.

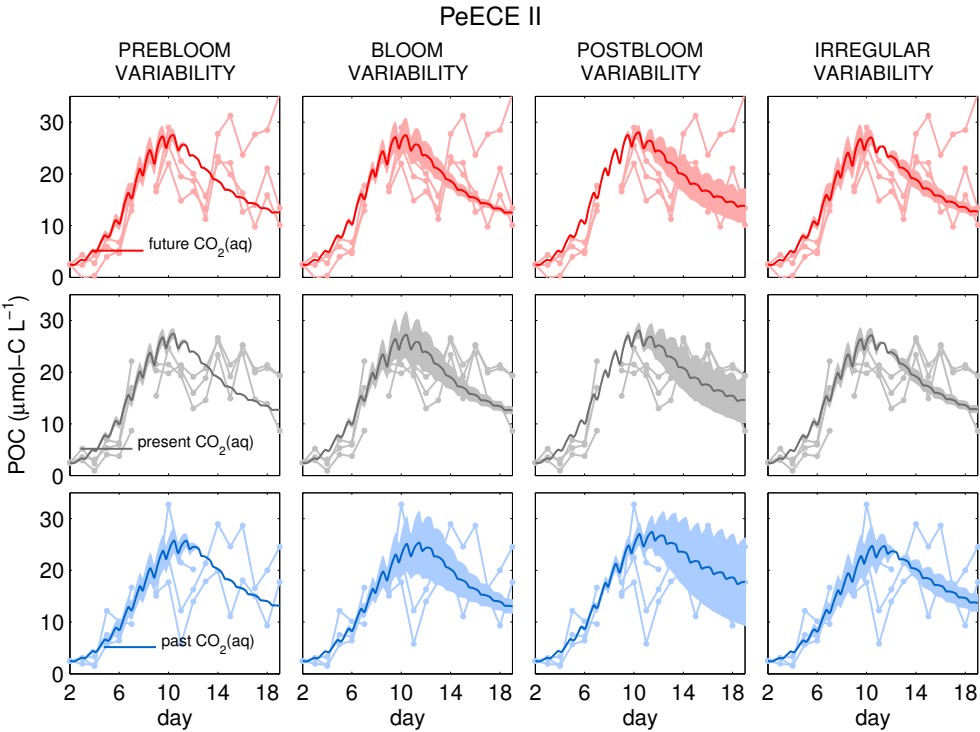

**Figure 5.** POC variability decomposition per factor, $\triangle POC_i^{mod}$ for PeECE II. Shaded areas are limited by the standard deviation of $10^4$ simulated POC time-series (see Sec. 2), around the mean trajectory of the ensemble (solid line). The timing of the amplification of the variability determines four separated kinds of behavior: factor uncertainties generating variability during the prebloom, bloom, postbloom or at irregular phase (see Sec. 3).

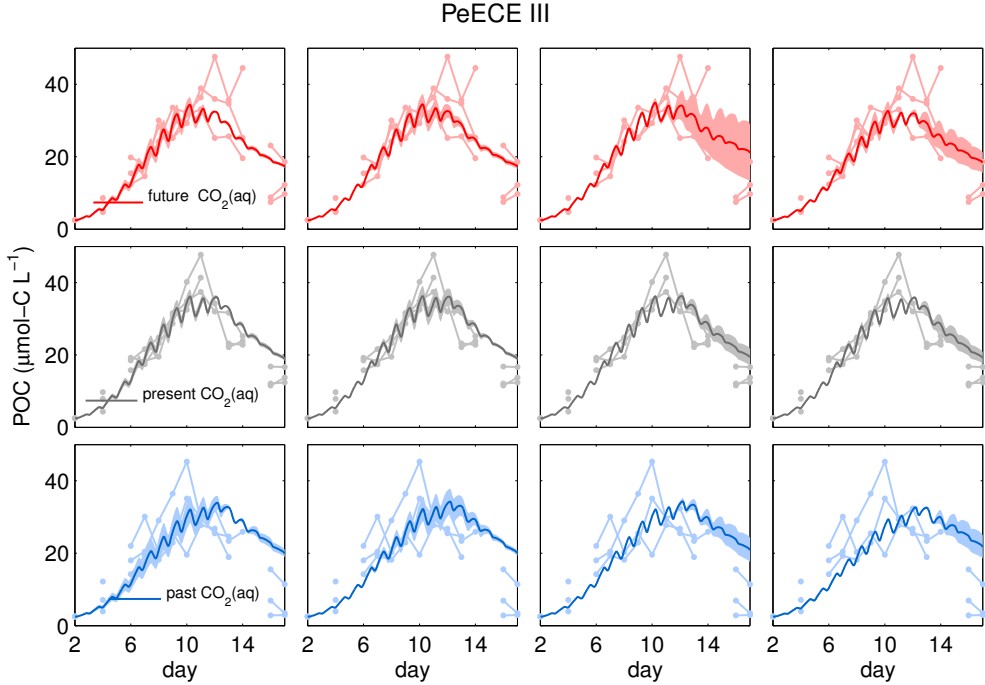

**Figure 6.** As Fig. 5, for PeECE III.

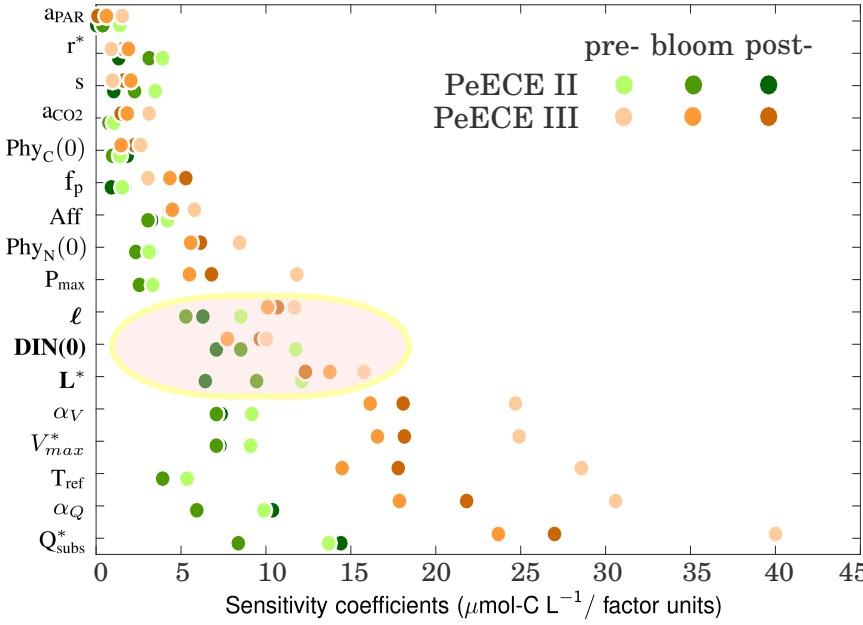

**Figure 7.** Sensitivity coefficients $\varepsilon_i$, Eq. (2) of factors $\phi_i$ listed in Tables 1 and 2 for different bloom phases in two OA independent mesocosm experiments. Factors whose uncertainties potentially mask acidification effects (Fig. 4) by triggering variability during the bloom (Figs. 5 and 6) are highlighted.

## Variability decomposition

### Experimental approach

### Model approach

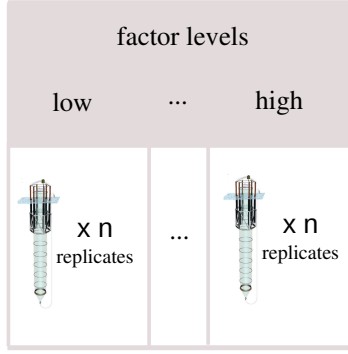

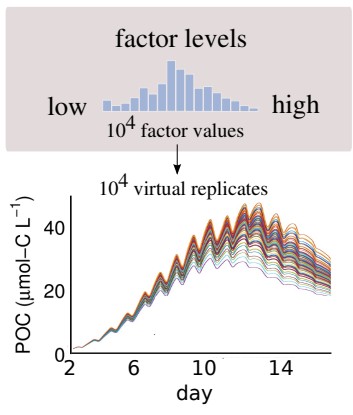

**Figure 8.** The exploration of the sources of variability in an experiment with a multi-way repeated measures ANOVA design with 3 acidification levels requires a multi-factorial high-dimensional setup (left panel). Alternately, we numerically simulate the biomass dynamics with $10^4$ virtual replicates, each one with a different normally distributed factor value (right panel). Uncertainty propagation relates the dispersion of the factor values with the dispersion of the POC trajectories. As an example, we plot results of POC variability in 50 virtual replicates of PeECE III at low acidification with uncertainty in initial nutrient concentration. Mesocosm drawing from Scheinin et al. (2015).

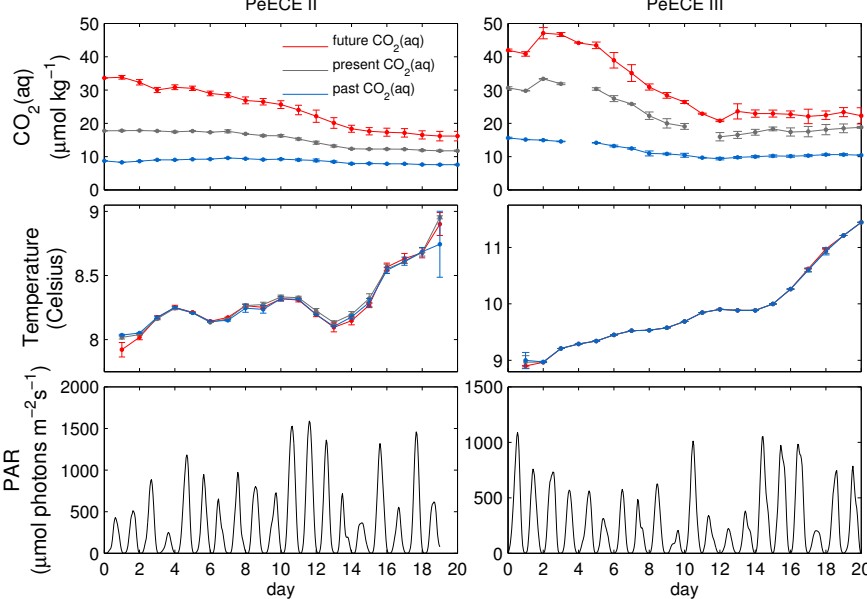

**Figure 9.** Environmental data from PeECE II and III are taken as model inputs. Error bars denote the standard deviation of the same treatment replicates.

**Table 1.** States variables and their dynamics.

| State variable | dynamical equation | ini. cond. | units |
|---|---|---|---|
| phytoplankton carbon | $\frac{dPhy_C}{dt} = (P\text{-}R\text{-}L) \cdot Phy_C$ | 2.5 | $\mu$mol-C L$^{-1}$ |
| phytoplankton nitrogen | $\frac{dPhy_N}{dt} = V \cdot Phy_C - L \cdot Phy_N$ | 0.4 | $\mu$mol-N L$^{-1}$ |
| nutrient concentration | $\frac{dDIN}{dt} = r \cdot DH_N - V \cdot Phy_C$ | 8$\pm$0.5 (*) | $\mu$mol-N L$^{-1}$ |
| | | 14 $\pm$ 2 (**) | $\mu$mol-N L$^{-1}$ |
| detritus and heterotrophs C | $\frac{dDH_C}{dt} = L \cdot Phy_C - (s \cdot DH_C + r) \cdot DH_C$ | 0.1 | $\mu$mol-C L$^{-1}$ |
| detritus and heterotrophs N | $\frac{dDH_N}{dt} = L \cdot Phy_N - (s \cdot DH_N + r) \cdot DH_N$ | 0.01 | $\mu$mol-N L$^{-1}$ |

* PeECE II, ** PeECE III

**Table 2.** Parameter values used for the reference run, $\langle \phi_i \rangle$. All values are common to both PeECE II and III experiments, only the mean temperature (determined by environmental forcing) and the averaged cell size in the community are different since different species composition succeeded in the experiments (*Emiliania huxleyi* was the major contributor to POC in PeECE II (Engel et al., 2008) but also diatoms significantly bloomed during PeECE III (Schulz et al., 2008).

| | Parameter | Value | Units | Variable | Reference |
|---|---|---|---|---|---|
| $a_{CO2}$ | carbon acquisition | 0.15 | $(\mu\text{mol-C})^{-1}\text{L}$ | $\text{Phy}_C$ | this study |
| $a_{PAR}$ | light absorption | 0.7 | $\mu\text{mol phot}^{-1}\text{m}^2$ | " | this study |
| $a^*$ | carboxylation depletion | 0.15 | $\mu\text{m}^{-1}$ | " | this study |
| $P_{max}$ | max. photosyn. rate | 12 | $\text{d}^{-1}$ | " | this study |
| $Q^*_{subs}$ | subsist. quota offset | 0.33 | $\text{mol-N}\,(\text{mol-C})^{-1}$ | " | this study |
| $\alpha_Q$ | $Q_{subs}$ allometry | 0.4 | - | " | this study |
| $\zeta$ | costs of N assimil. | 2 | $\text{mol-C}\,(\text{mol-N})^{-1}$ | " | Raven (1980) |
| $\ell$ | mean size Ln(ESD/$1\mu m$) | 1.6 | - | $\text{Phy}_C, \text{Phy}_N, \text{DIN}$ | PeECE II data |
| | | 1.8 | - | | PeECE III data |
| $f_p$ | fraction of protein in photosyn. machinery | 0.4 | - | " | this study |
| $V^*_{max}$ | max. nutrients uptake | 0.5 | $\text{mol-N}\,(\text{mol-C d})^{-1}$ | " | this study |
| Aff | nutrient affinity | 0.2 | $(\mu\text{mol-C d})^{-1}\text{L}$ | " | this study |
| $\alpha_V$ | $V_{max}$ allometry | 0.45 | - | " | Edwards et al. (2012) |
| $L^*$ | phyto. losses coeff. | $11 \cdot 10^{-3}$ | $(\mu\text{mol-C d})^{-1}$ | $\text{Phy}_C, \text{Phy}_N$ and $\text{DH}_C, \text{DH}_N$ | this study |
| $r^*$ | DIN remin. & excret. | 1.5 | $\text{d}^{-1}$ | $\text{DH}_C, \text{DH}_N$ | this study |
| s | DH sinking | 10 | $\text{L}(\mu\text{mol-C d})^{-1}$ | " | this study |
| $T_{ref}$ | referen. temperature | 8.3 | Celsius | $\text{Phy}_C, \text{Phy}_N$ and | PeECE II data |
| | | 10.1 | Celsius | $\text{DIN}, \text{DH}_C, \text{DH}_N$ | PeECE III data |

**Table 3.** Tolerance of mesocosms experiments to differences among replicates, given as a percentage of the reference factor value listed in Tables 1 and 2. According to our model projections, above these thresholds the simulated variability, $\triangle POC_i^{mod}$, exceeds the observed variability, $\triangle POC^{exp}$. Main contributors to the simulated variability during the bloom are highlighted in bold (see Sec. 3).

| factor $\phi_i$ | | \multicolumn{6}{c}{$\triangle\Phi_i$ (%)} | averaged |
| | | \multicolumn{3}{c}{PeECE II} | \multicolumn{3}{c}{PeECE III} | tolerance |
| | | Future | Present | Past | Future | Present | Past | (%) |
|---|---|---|---|---|---|---|---|---|
| $Phy_C(0)$ | initial phyto C biomass | 68 | 49 | 46 | 78 | 60 | 100 | $67 \pm 6$ |
| $Phy_N(0)$ | initial phyto N biomass | 26 | 19 | 22 | 21 | 16 | 29 | $22 \pm 4$ |
| **DIN(0)** | **initial DIN** | **20** | **28** | **29** | **17** | **11** | **18** | **$20 \pm 6$** |
| $a_{CO2}$ | carbon acquisition | 89 | 46 | 23 | 86 | 63 | 46 | $59 \pm 23$ |
| $a_{PAR}$ | light absorption | >100 | >100 | 98 | >100 | >100 | 92 | $> 100$ |
| $P_{max}$ | maximum photosyn. rate | 27 | 18 | 16 | 22 | 16 | 28 | $21 \pm 5$ |
| $Q_{subs}^*$ | subsistence quota offset | 6 | 5 | 6 | 5 | 4 | 9 | $6 \pm 1$ |
| $\alpha_Q$ | $Q_{subs}$ allometry | 9 | 7 | 8 | 7 | 5 | 10 | $8 \pm 2$ |
| $\ell$ | **size Ln(ESD/1$\mu$m)** | **25** | **20** | **29** | **19** | **14** | **22** | **$22 \pm 5$** |
| $f_p$ | fraction of protein in photosyn. machinery | 92 | 75 | 44 | 36 | 17 | 38 | $50 \pm 25$ |
| $V_{max}^*$ | maximum nutrient uptake | 13 | 11 | 14 | 10 | 8 | 14 | $12 \pm 2$ |
| Aff | nutrients affinity | 39 | 31 | 42 | 38 | 36 | 55 | $40 \pm 7$ |
| $\alpha_V$ | $V_{max}$ allometry | 14 | 11 | 15 | 10 | 8 | 14 | $12 \pm 2$ |
| $L^*$ | **phytoplankton losses** | **22** | **30** | **28** | **12** | **10** | **15** | **$20 \pm 8$** |
| $r^*$ | DIN remineralization | 73 | 99 | 98 | 128 | 37 | 52 | $81 \pm 31$ |
| s | DH sinking | $> 100$ | $> 100$ | 96 | $> 100$ | 61 | 79 | >100 |
| $T_{ref}$ | reference temperature | 17 | 12 | 14 | 9 | 7 | 14 | $12 \pm 3$ |

Discussion Paper | Discussion Paper | Discussion Paper | Discussion Paper |

**Table 4.** Cumulative residuals for PeECE III.

| Y | E | M | units |
|---|---|---|---|
| POC | 35.1 | 37.4 | $\mu$mol-C L$^{-1}$ |
| PON | 6.0 | 9.1 | $\mu$mol-N L$^{-1}$ |
| DIN | 6.7 | 9.2 | $\mu$mol-N L$^{-1}$ |