# Peer review of "Potential sources of variability in mesocosm experiments on the response of phytoplankton to ocean acidification"

_Biogeosciences, 2016_

## Referee Comment (RC1) · Anonymous Referee #1 · 8 Jun 2016

This is an interesting study which uses a modelling approach to (potentially) explore sources of the variability observed in mesocosm studies. Specifically, in two such studies which have investigated the impacts of ocean acidification on phytoplankton communities, and associated production of particulate organic matter (POM); specifically, particulate organic carbon (POC). I think the study is very interesting model perturbation exercise, in that it explores how specific parts of the model impact on variability in the production of POC, which could be potentially useful to the climate change and ocean acidification community. However, I have important concerns about how this is sold throughout, and believe that the inferences made about how these results may help explain variability observed in mesocosm experiments are not entirely supported by the study design used. Specifically, because of the procedures used to produce the simulations, as detailed below. I therefore find it difficult to accept this study for

publication in its present form and would suggest that to that aim, major revisions are needed addressing the primary point raised below. I am uncomfortable with the assertion that these model simulations porduced in this study can be used to define which uncertainties lead to observed variably in experimental results (L4-6, page 1). In your model, a number of mechanisms is represented, with associated parameter spaces investigated here. The optimization (as far as I understand it) of this parameter set is based on the POC observations in the experiments, and given (as far as I can gather) that POC is an emergent property of your model, your modelled POC can therefore not be used to infer causality between these mechanisms in the model and POC in the experiments. Indeed, in page 4, L29-32, it seems you optimize the parameter values by minimizing a model costs based on an emergent property (POC) (i.e. the relationship between POCexp and POCmod), not your state variables. Is this the case? If so, you cannot use this study to infer about what is observed in POC in the experiments, because you cannot trace how likely your parameter values used, or the simulations of your state variables are. You do no use any observational datasets to validate the state variables simulations associated with the mechanisms represented, or the parameter values chosen here (adjusted?)? Through the various optimizations of parameter values, it is therefore possible that random combinations of parameters can be achieved to produce the observed variability in POC, or close to it, but this does not mean that this is what happens in reality. I believe you address this partially at the end of the methods section (L24-27, page 5), but isn't this the rationale of your study? If so, then your study is not what is advertised in the title and abstract. What you do here is explore how a set of mechanisms represented in your model may produce a progression of POC values. It is a model perturbation exercise. But you have tuned the parameters through the simulation so that your result is close to the observations – is this true? You further tune POCmod in a specific way (using an assumed relationship) because you did not include other elements of the ecosystem contributing to POC in the experiments (Appendix D, page 12). So you cannot then infer that mechanisms in your model are those affecting POC in the mesocosm experiments. This is particularly important given that you assume that increased DIC increases primary production (L27 onwards, page 3) but this is not a ubiquitous perspective in the community (e.g. Artioli et al 2014 Biogeosciences Discussions 11: 601-612; Nagelkerken & Connel 2015 PNAS 112: 13272–13277). To do so, to investigate this causality between the mechanisms in your model and POC in the experiments, I suppose you could produce POC simulations with different parameter values without tuning them. Or you could use experimentally derived parameter values for those changing conditions and see the effect on modelled POC, in different scenarios. In these two cases you could then investigate how changes to those parameters, and the mechanisms in your model that they affect, impact on the outcome (POC). To accept this work for publication, I assume that the authors would have to explain why they disagree with my reading of their work, or they would have to requalify the assertions made throughout about the implications of their findings along these lines. Detailed comments: Text could benefit from being edited by a native English speaker, and I have not undertaken those corrections. Introduction Line 18 page 1: particular organic matter = POM; particulate organic carbon = POC. Please define all acronyms on first use, throughout. L17-20, page 2: Sections unclear. Please rephrase without jargon. L24: "alternative" is not the best word here. "Complimentary" would be more adequate. L24 page 1: sensitivity (analysis), not robustness. Method Line 18 onwards, page 3: please define all acronyms used at first mention, here and in tables. L17-18, page 4: does "adjusted" describe tuning? Are there no parameters estimated from the experiments that may be used to validate/address the set of parameter values used in the various simulations? I.e. are all parameters effectively tuned to acquire the best agreement between simulated POC and observed POC? Page 4: reference is made to the use of normal distribution to produce variation in model factors. I suppose you mean in the model parameters but not in the state variables (in page 3, L10 you say model factors are the initial parameter values and the initial conditions of the state variables). Did you constrain the normal distributions sampled in order to limit the parameter values to positive values? If so, please state this and explain how you constrained the sampled distributions used for delta phi. Perhaps I am missing something here? Could you explicitly provide the values you used in the definition of future, present and past CO2 conditions? Could you provide the references you used to define your initial conditions for all your model factors (Tables I and II)? My expertise in plankton ecology is not sufficient to allow me to comment on the actual, values used as initial conditions for the model. So these should be reviewed by someone with that knowledge.

L3-5 page 5: is this a reasonable expectation, given that your parameters are not independent? E.g. aCO2 is possibly quite tightly dependent on V*max? Please expand on why you think this is an appropriate assumption. Results L7-10 page 6: it could be argued that the purpose of conducting mesocosm experiments in real life (usually to investigate what we think are the mechanism underlying variation in some variable in real life) is to observe what mean and variability we get under pre-determined conditions. To bind the initial conditions of the experiment in order to modify the result (variability) could be perceived to be a circular argument. L14-onwards: if the parameters are tuned based on model cost calculated using POCexp, how can we be sure that this matters in any way other than in the model structure used here? What you have carried out is a model perturbation experiment, with tuning of parameters. I find it difficult to determine how we can derive new knowledge about the way in which ocean acidification impacts plankton communities. Discussion L1-8: please requalify these sentences (significantly) based on main concern raised above. I found it difficult to comment on the discussion given my major concern above. I believe this is an interesting approach but that the inferences made in this section (sources of variability of POC in experimental datasets) are not supported by the study design used. Conclusion L9-11 page 9: I think that you potentially miss-sell the importance of your study. Manipulating the aspects you suggest to affect experiment results seems circular. However, you have a very interesting model with which to explore how climate change and ocean acidification impact phytoplankton dynamics, which could be very useful to the community. This is a very interesting tool for ecologists to test the influence of specific mechanisms represented in your models. Future validation of this approach through use of potentially different approaches for parameter optimization and use of observational data to evaluate model skill would be important.

---

## Referee Comment (RC2) · Anonymous Referee #2 · 24 Aug 2016

General comments Variability in responses and interpretation of the results of acidification experiments is a relevant problem and the topic of this paper is interesting. However the title does not accurately reflect the content of the paper, as it is general enough to have a reader believe the article treats variability in all kinds and types of mesocosm experiments using acidification treatments while in fact the focus of the paper is an in depth analysis of the cause of variability in two specific experiments and only dealing with primary producer responses.

The paper is well structured and clear and easy to follow.

Specific comments The modeling approach described is interesting and novel as a tool for investigating the source of variability in acidification experiments. With this approach the authors prove that variability in initial conditions can generate the observed

variability and specify which are important to maintain similar as starting values (nutrient concentration, cell size and biomass in this case). Interesting as this is, this might be very difficult to control in field-based experiment where the environment is naturally patchy while increasing the number of replicas is usually not an option either.

Useful suggestions for the interpretation of effects of OA are evaluating differences between the slope of the growth phase and a complete characterization of phytoplankton biomass loss. However, these suggestions are limited to OA experiments with primary producers.

As indicated on page 15, line 13-14 the described model should be seen as a starting point and can be applied to any pertubation experiment with highly variable responses: "With this study we established a foundation for further model-based analysis for uncertainties propagation that can be generalized to any kind of experiments in biogeoscience." The described modelling exercise on primary producer responses to OA in a specific mesocosm set-up is interesting but should be presented as such with an appropriate title.

Technical corrections

Introduction Line 5 – sever should be severe line 9 – mesocosm should be plural (mesocosms as in line 7)

Method Line 7 – not coupling should be no coupling?

Discussion Page 11, line 25 – bound should be bind

---

## Author Comment (AC1) · 1 Nov 2016

We thank Referee #1 for providing helpful comments and we greatly appreciate the time and effort Referee #1 had spent to review our manuscript. Most of the Referee's comments were considered and we introduced major changes to our manuscript. By following the Referee's comments we could identify and remove many ambiguities. In particular, we eliminate the term 'sensitivity analysis'. That method (typically used to increase understanding of the relationships between input and output variables for model simplification or error identification) has some similarities and can be easily mistaken with the uncertainty quantification method we apply here. Besides, we learned that the term 'effect sizes' is too ambiguous in the context of our study. We thus eliminated it in favor of 'sensitivity coefficients'.

[Figure]

In the revised version of our manuscript we have further clarified the purpose of our analysis approach. With our approach we aim at providing support to experimental work in our field of science by specifying potential sources of observed variability. As pointed out by Referee #1, our study on variability decomposition depends on the underlying model assumptions. In spite of its simplicity, the model yields results that explain data from independent experiments, which confirms its applicability. Moreover, our approach goes beyond those analyses that typically focus on "mean dynamics" by also simulating the data variability. In order to make our method more comprehensive we prepared a new figure that compares our mathematical approach with the equivalent experimental setup (mesocosm drawings from Scheinin et al. 2015, J. R. Soc, Interface 12:20150056). The presented approach is, to our knowledge, the first applied for analyzing complex data from mesocosm experiments. For further clarification, we have also included a conceptual diagram in the revised method section.

In the following we provide our responses to individual comments raised by of Referee #1.

Comment 1 (C1): "I am uncomfortable with the assertion that these model simulations produced in this study can be used to define which uncertainties lead to observed variably in experimental results (L4-6, page 1)"

Authors' response to C1:

The model is a mechanistic description of plankton growth dynamics based on dynamically and ecologically consistent equations. Due to the skill of the simple model and the consistency of our methodology, we regard our results as reasonable estimates of how the investigated factors can generate the observed variability among replicates. However, we decided to address this concern of Referee #1 and explicitly mention the caveat that an agreement between experimental data and simulation results does not necessarily imply uniqueness and maximum of realism reproduced by the model structure and by the parameterizations imposed. As a consequence, results based on any

particular model are always prone to indetermination.

Regarding the variability decomposition method, we performed uncertainty propagation following the "Guide to the expression of uncertainty in measurement" (GUM), in particular, the Supplement 1: "Propagation of distributions using a Monte Carlo method", prepared in 2008 by the international Joint Committee for Guides in Metrology (JCGM). This method has been adopted by many organizations, is widely used, and has been implemented in standards and guides on measurement uncertainty.

Comment 2 (C2): "...  you cannot use this study to infer about what is observed in POC in the experiments, because you cannot trace how likely your parameter values used, or the simulations of your state variables are. You do not use any observational datasets to validate the state variables simulations associated with the mechanisms represented, or the parameter values chosen here (adjusted?)?

Authors' response to C2:

The reference solution was obtained by adjusting parameter values so that they can be used for simulations of two independent experiments. If we had applied a data assimilation approach we may have identified parameter values based on probabilistic considerations, e.g. by maximizing a likelihood. Doing so does not automatically provide reasonable parameter estimates. The additional application of a data assimilation approach is clearly out of scope of the study presented here. The 'adjustment' of the parameter values to provide a qualitative fit to experimental data is rather common than unusual. In preparation of our study described here we performed an extensive series of simulation runs, with models of different complexity and with different parameterizations. This preparatory work is not documented explicitly, as it does not provide much insight with respect to the objective of the manuscript. For the same reason we did not introduce a data assimilation method for specifying a reference model solution.

The major achievement during the process of model selection and of parameter adjustment was to identify the simplest model structure while keeping parameter values

within meaningful limits. Thus, we have identified an effective representation of major processes underlying the observed experimental dynamics, which specified the model reference solutions. This 'calibrated' reference solution successfully explains 54 data sets of repeated measures of POC, PON and DIN (i.e., 3 quantities x 3 treatment levels x 3 replicates x 2 independent experiments on ocean acidification with primary producers).

We added some more information (second paragraph section 2.1) about the calibration of the model and how the reference solution is identified. In fact, we also considered a third independent data set for model validation. The third data set is not included since it is not directly relevant for the study presented here.

Comment 3 (C3): "Through the various optimizations of parameter values, it is therefore possible that random combinations of parameters can be achieved to produce the observed variability in POC, or close to it, but this does not mean that this is what happens in reality. I believe you address this partially at the end of the methods section (L24-27, page 5), but isn't this the rationale of your study?"

Authors' response to C3:

We disagree with the Referee's remark that any random combination of parameter values would yield similar variability on POC. We recall that no systematic data assimilation method was employed for parameter optimization. By 'optimization of parameter values' we assume that Referee #1 actually refers to the iterative procedure of assessing the limits of variations of parameter values that generate variations in model states at specific times. These variations in model states (like in POC) are thus mechanistically (dynamically) linked to the variations of parameter values, including initial conditions. The parameter values were varied individually while other parameter values remained fixed (note that the fixed values correspond with the calibrated model reference solution).

In L24-27, page 5, we noticed that a substantial achievement already is to mechanistically elucidate a minimum number of requirements that are sufficient for the uncertainty to escalate and mask treatment effects (specially since it is not possible to find the total number of requirements, neither experimentally nor mathematically). The rationale of our study is to show that slight differences in initial conditions (in particular, in nutrient concentration, mean cell size or biomass losses) are already sufficient to blur the signal of treatment effects.

Comment 4 (C4): "Indeed, in page 4, L29-32, it seems you optimize the parameter values by minimizing a model costs based on an emergent property (POC) (i.e. the relationship between POCexp and POCmod), not your state variables. Is this the case?"

Authors' response to C4:

We learned that the essence of the method is not sufficiently well documented. The Referee's comment is helpful. A conceptual diagram of the method has been devised and included to our revised manuscript, see new Fig. (1). The diagram illustrates all steps of the work flow of the analysis: the model was calibrated with POC, PON and DIN experimental data (calculation of the reference run, steps 1 and 2 in Fig. 1); later, we perform the uncertainty propagation analysis (part of it is the comparison of POC experimental and simulated variability to estimate the tolerance thresholds, step 5 in Fig. 1).

The model counterpart to the POC measurements is equal to the sum of two state variables, namely phytoplankton carbon and the carbon pool attributed to detritus and all heterotrophs: POC = PhyC + DH_C (see former L23, page 3).

Comment 5 (C5): "... you assume that increased DIC increases primary production (L27 onwards, page 3) but this is not a ubiquitous perspective in the community (e.g. Artioli et al 2014 Biogeosciences Discussions 11: 601-612; Nagelkerken & Connel 2015 PNAS 112: 13272–13277)."

Authors' response to C5:

We find sufficient evidence in the literature that primary production becomes enhanced under elevated CO2 conditions. This enhancement may seem to be low and whether it can be unambiguously revealed depends on the experimental design. We also follow theoretical considerations as described in Wirtz (2011, Journal of Phytoplankton Research 33, 9:1325-1341, that are in support of finding enhanced carbon fixation rates with increasing CO2 levels. We greatly appreciate that Referee #1 provided two valuable references. In Artioli et al. (2016, Biogeosciences Discussions 11: 601-612) it is straightened that high CO2 enhances primary production (and in that article also PeECE III data are used). In the second study, by Nagelkerken & Connel (2015, PNAS 112: 13272–13277), it is shown that high CO2 enhanced primary production in most cases while the variability in the data may become too large. These studies are good examples that highlight the relevance of our study: controversial results in ocean acidification experiments. We have considered both studies in the Introduction and Methods sections.

Responses to detailed comments by Referee #1

Detailed comment 1 (DC1): "Did you constrain the normal distributions sampled in order to limit the parameter values to positive values? If so, please state this and explain how you constrained the sampled distributions used for delta phi"

Authors' response to DC1:

We thank Referee #1 for this notice. We added a sentence at the end of the Methods section to explain that we dismissed negative values, representing less than 5% of all trajectories. Given the super-optimal number of virtual replicates, this reduction did not affect the results.

Detailed comment 2 (DC2): "Could you explicitly provide the values you used in the definition of future, present and past CO2 conditions? Could you provide the references you used to define your initial conditions for all your model factors (Tables I and II)? My expertise in plankton ecology is not sufficient to allow me to comment on the

actual, values used as initial conditions for the model. So these should be reviewed by someone with that knowledge."

Authors' response to DC2:

The CO2 values we used as forcing were plotted in Appendix D. They were downloaded from PANGAEA, together with the initial conditions of the state variables given in Table (1). Table (2) lists the parameter values used in the reference run. Many parameters are difficult to measure and have not been experimentally determined, but the values used remain within the range of plausible biological values.

Detailed comment 3 (DC3): "L3-5 page 5: is this a reasonable expectation, given that your parameters are not independent? E.g. aCO2 is possibly quite tightly dependent on V*max? Please expand on why you think this is an appropriate assumption."

Authors' response to DC3:

We refer to independence of errors (i.e. covariances being zero), not the independence of the parameters (no collinearities). The independence of the uncertainties is extensively assumed (we have included the reference to the GUM). Although this is a typical assumption, we agree that this assumption introduces limitations and the consideration of correlations will likely be an improvement.

Our model accounts for the dependence between aCO2 and V*max because we follow Edwards et al. (2011, Ecology 92:2085–2095) and resolve allometric relationships that describe the relation between maximum growth and nutrient uptake through the logarithm of the equivalent spherical diameter (GUM Supplement 1, Section 6.1.4 NOTE: "It may be possible to remove some or all dependencies by re-expressing relevant input quantities in terms of more fundamental independent input quantities on which the original input quantities depend").

Detailed comment 4 (DC4): "Results L7-10 page 6: it could be argued that the purpose of conducting mesocosm experiments in real life (usually to investigate what we think

are the mechanism underlying variation in some variable in real life) is to observe what mean and variability we get under pre-determined conditions. To bind the initial conditions of the experiment in order to modify the result (variability) could be perceived to be a circular argument."

Authors' response to DC4:

Effects of the treatment are expected to appear as differences among treatment levels, not as differences among replicates of the same treatment level. We provide an estimate of how much the latter need to be constrained in order to observe the former. We do not suggest to 'bind' the initial conditions of different treatment levels since the exploration of the differences among treatment levels is the aim of the experiment, as correctly pointed by Referee #1.

Detailed comment 5 (DC5): "L14-onwards: if the parameters are tuned based on model cost calculated using POCexp, how can we be sure that this matters in any way other than in the model structure used here? What you have carried out is a model perturbation experiment, with tuning of parameters. I find it difficult to determine how we can derive new knowledge about the way in which ocean acidification impacts plankton communities."

Authors' response to DC5:

The main point is that our method involves a mechanistic understanding of how the plankton community can react to ocean acidification. Based on the available data and on this mechanistic description we make inferences about $CO_2$ effects on the timing and intensity of the phytoplankton bloom (it is earlier and larger under high $CO_2$ conditions) and about the origins of variability in observations from ocean acidification mesocosm experiments that include a natural plankton community. Furthermore, we can disentangle differences in observed POC in response to a $CO_2$ effect and in response to variations in ecophysiological factors ($phi_i$). To make similar inferences from statistical analyses of the data is hardly possible, unless such analysis accounts

for some of the predominant interdependencies between nonlinear processes. The mechanistic model description introduces an explicit representation of such nonlinearities.

We thank Referee #1 for sharing her/his thoughts and for proposing two new references. The comments helped to improve the manuscript and to avoid any further misconceptions. Following her/his suggestion, we hired an English editing service for the manuscript text as well.
* * *
[Figure]

**Fig. 1.** Conceptual diagram for the variability decomposition method based on uncertainty propagation

**Variability decomposition**

**Experimental approach**

[Figure]

factor levels

low ··· high

x n replicates ··· x n replicates

x N factors
x 3 acidification levels

**Model approach**

[Figure]

factor levels

low high

$10^4$ factor values

$10^4$ virtual replicates

POC ($\mu$mol–C L$^{-1}$)

x 19 factors
x 3 acidification levels

**Fig. 2.** The exploration of the sources of variability in an experiment requires a multi-factorial high-dimensional set-up (left). Alternately, we simulate the biomass dynamics with virtual replicates (right)

---

## Author Comment (AC2) · 1 Nov 2016

We thank Referee #2 for the valuable comments on our manuscript. We agreed in all aspects and implemented all the suggested corrections. Major changes have been introduced to our manuscript. In particular, we renamed the manuscript as "Potential sources of variability in mesocosm experiments on phytoplankton response to ocean acidification" to accurately reflect the content of the paper. We think that the revised manuscript is more coherent and clearer with respect to our approach and our results. Many refinements in our manuscript involve the validity of the method, also inspired by comments of Referee #1.

---

## Author Response (AR2)

Author's response to comments on version 5

Minor comments by the referee:

**Comment 1**:
I would recommend that you define more clearly the meaning in your paper of (ecophysiological) "uncertainty" (uncertainty is defined on page 3, line 28 but I suggest that you clarify that, in this paper, this relates to the random variations in the initial conditions of the experiments (if I understood correctly), "variability", "sample/replicate", factors, ..already in the Introduction. I agree that some of these terms are defined afterwards in the material and methods section but some specifications of their meaning in this paper would be helpful in order to understand the introduction and, in particular, the objectives on of the study.

**Authors' response**: We agree and see that an explanation of the terms "uncertainty", "variability", "replicates", "factors" already in the introduction is meaningful. We decided to shorten and revise text in the introduction section (Sect. 1). Doing so, we realised that the original paragraph (page 4, line 5 to page 5, line 4): "The confirmation... " should rather be placed in the discussion section. We included minor changes to this paragraph and moved it to the discussion section (Sect. 4). The respective paragraph was assigned to a new subsection (Sect. 4.4) with the title "Inference from summary statistics on mesocosm data".  Overall, with the revised introduction section we hope to have clarified the meaning of "variability", "replicate", "factors" and "uncertainties" in the context of our study. We are thankful for the referee's suggestion, as we think that we could further improve readability and provide a better access to the content of our study.

**Comment 2**:
Indeed, I find that, as it is now, the description of the objectives of the manuscript (page 5, lines 4-16) is still not straightforward to understand especially for a general reader, although these lines are important as they define the scope of the paper. For instance, you mentioned: "In this study, we estimate the effects of ecophysiological uncertainties by associating the variability in experimental observations to a variational range from repeated model runs". This sentence is very technical (especially its second part when you are not an expert and when you have not read the whole manuscript). I would suggest some reformulations. For instance (but it can be of course different), *"We estimate the effects of ecophysiological uncertainties (i.e; different initial conditions in a selected number of factors ...) on the variability in experimental observation (here the POC variations). With that aim an ensemble of model simulations starting from a range of selected factors is*

*performed. The range of selected factors is defined so as the variability (i.e. standard deviation) of model outputs does not exceed that of experimental data during the course of the experiment".*

**Authors' response**: We understand the referee's concern and we addressed this while revising the introduction.  The example sentence provided by the referee is very helpful in this respect. We included to the last paragraph of our introduction section with the following sentences: "The central idea is to produce ensembles of model simulations, starting from a range of values for selected factors. The range of values, for selected factors, is determined so as the variability in model outputs does not exceed variability in observations over the course of the experiment."

**Non-public comments to the Author**:
As you have seen the reviewer is still not really convince that the outputs of the study is really dependent on the model used and this limits its scope of application. He/she still rated the "Scientific significance         of         the         study"         as         "fair". Also, I would like to encourage you that you add some considerations on how your conclusions may depend on the model structure used in this paper but why you believe that nevertheless the messages are robust and important to consider for the design of future mesocosms studies. The recommendations for the preparation of mesocosms experiments is probably an important objective of this manuscript.

**Authors' response**: We greatly appreciate patience and support of the editor. We sought to follow her recommendation by adding a paragraph about the model design and the expected outcome if some other model was employed. We again stress that given the similarity of our model to many other state-of-the-art descriptions of phytoplankton growth and owing to (not shown) analyses of predecessor versions of the model, outcomes will be at least in part transferable. If we would further increase complexity by dealing with several model structures, this would likely deteriorate accessibility of the paper. We however not fully comply with the evaluation of the work as  being "fairly significant "  since it introduces a novel model-based approach for dealing with experimental uncertainties, which we think are still undervalued in life sciences.

This additional paragraph reads as follows:
"The results of our analyses are conditioned by the dynamical model equations imposed. Deliberately, the model's complexity is kept low, mainly to limit the generation of structural errors with respect to model design. At the same time, the level of complexity resolved by the model suffices to explain POC measurements of two

independent mesocosm experiments with identical parameter values (see Table 2), which highlights model skill. The equations used comply with theories of phytoplankton growth (e.g. Droop, 1973; Aksnes and Egge, 1992; Pahlow, 2005; Edwards et al., 2007; Litchman et al., 2007; Wirtz, 2011). The uncertainty propagation employed here can be applied to any model. As long as the model features a similar structural complexity and is also able to reproduce POC with sufficient accuracy, we expect similar qualitative findings with respect to the factors ($\Phi_i$) and the identification of the major contributors to the variability. However, we would not expect other models to reveal similar values in the ratio $\epsilon_i$, which would likely depend on the equations used to resolve some of the ecophysiological details."

Author's response to comments on version 5

Minor comments by the referee:

**Comment 1**:
I would recommend that you define more clearly the meaning in your paper of (ecophysiological) "uncertainty" (uncertainty is defined on page 3, line 28 but I suggest that you clarify that, in this paper, this relates to the random variations in the initial conditions of the experiments (if I understood correctly), "variability", "sample/replicate", factors, ..already in the Introduction. I agree that some of these terms are defined afterwards in the material and methods section but some specifications of their meaning in this paper would be helpful in order to understand the introduction and, in particular, the objectives on of the study.

**Authors' response**: We agree and see that an explanation of the terms "uncertainty", "variability", "replicates", "factors" already in the introduction is meaningful. We decided to shorten and revise text in the introduction section (Sect. 1). Doing so, we realised that the original paragraph (page 4, line 5 to page 5, line 4): "The confirmation... " should rather be placed in the discussion section. We included minor changes to this paragraph and moved it to the discussion section (Sect. 4). The respective paragraph was assigned to a new subsection (Sect. 4.4) with the title "Inference from summary statistics on mesocosm data". Overall, with the revised introduction section we hope to have clarified the meaning of "variability", "replicate", "factors" and "uncertainties" in the context of our study. We are thankful for the referee's suggestion, as we think that we could further improve readability and provide a better access to the content of our study.

**Comment 2**:
Indeed, I find that, as it is now, the description of the objectives of the manuscript (page 5, lines 4-16) is still not straightforward to understand especially for a general reader, although these lines are important as they define the scope of the paper. For instance, you mentioned: "In this study, we estimate the effects of ecophysiological uncertainties by associating the variability in experimental observations to a variational range from repeated model runs". This sentence is very technical (especially its second part when you are not an expert and when you have not read the whole manuscript). I would suggest some reformulations. For instance (but it can be of course different), *"We estimate the effects of ecophysiological uncertainties (i.e; different initial conditions in a selected number of factors ...) on the variability in experimental observation (here the POC variations). With that aim an ensemble of model simulations starting from a range of selected factors is*

*performed. The range of selected factors is defined so as the variability (i.e. standard deviation) of model outputs does not exceed that of experimental data during the course of the experiment".*

**Authors' response**: We understand the referee's concern and we addressed this while revising the introduction.  The example sentence provided by the referee is very helpful in this respect. We included to the last paragraph of our introduction section with the following sentences: "The central idea is to produce ensembles of model simulations, starting from a range of values for selected factors. The range of values, for selected factors, is determined so as the variability in model outputs does not exceed variability in observations over the course of the experiment."

**Non-public comments to the Author**:
As you have seen the reviewer is still not really convince that the outputs of the study is really dependent on the model used and this limits its scope of application. He/she still rated the "Scientific significance        of        the        study"        as        "fair". Also, I would like to encourage you that you add some considerations on how your conclusions may depend on the model structure used in this paper but why you believe that nevertheless the messages are robust and important to consider for the design of future mesocosms studies. The recommendations for the preparation of mesocosms experiments is probably an important objective of this manuscript.

**Authors' response**: We greatly appreciate patience and support of the editor. We sought to follow her recommendation by adding a paragraph about the model design and the expected outcome if some other model was employed. We again stress that given the similarity of our model to many other state-of-the-art descriptions of phytoplankton growth and owing to (not shown) analyses of predecessor versions of the model, outcomes will be at least in part transferable. If we would further increase complexity by dealing with several model structures, this would likely deteriorate accessibility of the paper. We however not fully comply with the evaluation of the work as  being "fairly significant "  since it introduces a novel model-based approach for dealing with experimental uncertainties, which we think are still undervalued in life sciences.

This additional paragraph reads as follows:

[revised manuscript text omitted]

**List of changes**

Paragraph move from Introduction to Discussion (new subsection 4.4).